# Discovering Compositional Hallucinations in LVLMs

**Sibei Yang**[1][†]   **Ge Zheng**[2]   **Jiajin Tang**[2]   **Jiaye Qian**[2]   **Hanzhuo Huang**[2]   **Cheng Shi**[3]

[1]School of Computer Science and Engineering, Sun Yat-sen University
[2]ShanghaiTech University   [3]School of Computing and Data Science, The University of Hong Kong

## Abstract

Large language models (LLMs) and vision-language models (LVLMs) have driven the paradigm shift towards general-purpose foundation models. However, both of them are prone to hallucinations, which compromise their factual accuracy and reliability. While existing research primarily focuses on isolated textual- or visual-centric errors, a critical yet underexplored phenomenon persists in LVLMs: Even neither of textual- or visual centric errors occur, LVLMs often struggle with a new and subtle hallucination mode that arising from composition of them. In this paper, we define this issue as Simple Compositional Hallucination (SCHall). Through an preliminary analysis, we present two key findings: (1) visual abstraction fails under compositional questioning, and (2) visual inputs induce degradation in language processing, leading to hallucinations. To facilitate future research on this phenomenon, we introduce a custom benchmark, SCBench, and propose a novel VLR-distillation method, which serves as the first baseline to effectively mitigate SCHall. Furthermore, experiment results on publicly available benchmarks, including both hallucination-specific and general-purpose ones, demonstrate the effectiveness of our VLR-distillation method.

## 1   Introduction

Large language models (LLMs) [3, 62, 8] and large vision-language models (LVLMs) [2, 5, 7, 37, 78, 4, 9] have driven the paradigm shift from task-specific to general-purpose approaches, cementing their role as the *de-facto* foundation in natural language processing and computer vision research. However, both LLMs and LVLMs are prone to hallucinations [24, 73, 55, 77, 15], posing significant risks in real-world applications [74, 68]. In LLMs, hallucination research primarily addresses discrepancies between model outputs and real-world facts or user inputs—*i.e.*, factuality hallucination and faithfulness hallucination [21]. Compared to LLMs, LVLMs incorporate visual understanding, which naturally extends hallucinations to include visual recognition errors—textual responses inconsistent with the referenced image—particularly in object categories [55, 33, 77], attributes, and relationships [64, 27]. To suppress these hallucinations, recent work has achieved promising results through improved architecture [61, 7], inference interventions [29, 22, 66, 28, 18], and auxiliary training data or strategies [77, 26, 72, 52, 76].

Despite recent progress, most existing studies [33, 55, 34, 11] focus on isolated forms of hallucination—either textual factuality and faithfulness errors or visual recognition failures (see POPE [33] and TruthfulQA [34] in Figure 1a). *But what if neither occurs on its own?* Intuitively, if an LVLM answers both a simple vision-centric and a simple language-centric question correctly, without hallucination, it should also succeed when the two are composed into a single query. *Yet, unexpectedly, it hallucinates.* We observe that when these seemingly reliable components are combined into a single question, the LVLM fails—hallucinating where no error existed before. For example, as shown in Figure 1(b), the LVLM independently recognizes the goldfish in the image and understands that adding more

---

[†]Corresponding author is Sibei Yang.

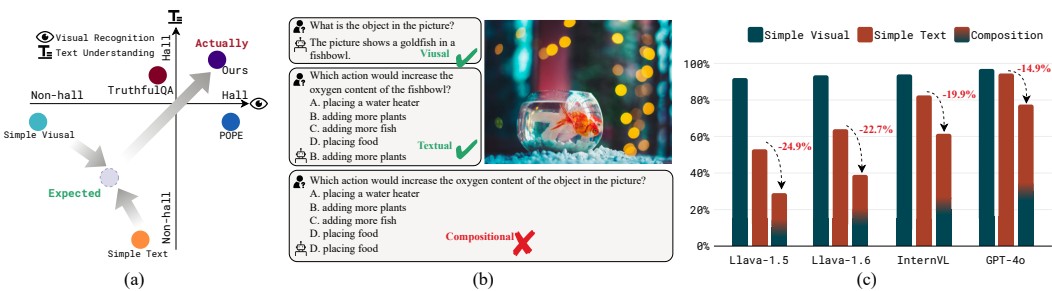

Figure 1: (a) Comparison of the research scope of our SCHall with prior work. (b) Example of SCHall. LVLMs provide accurate answers to simple visual- or textual-centric questions but fail to reason compositionally when these questions are combined. (c) LVLMs exhibit SCHall on our compositional benchmark, showing an performance drop of 20% compared to individual questions.

plants increases the oxygen content of the fishbowl. However, when asked the compositional question "which action would increase the oxygen content of the object", it hallucinates and produces an incorrect answer.

In this paper, we define this phenomenon as **Simple Compositional Hallucination (SCHall)**: *a new and subtle hallucination mode that does not arise from textual- or visual-centric questions individually, but from their compositions—particularly when each component question is simple and independently hallucination-free.* To better investigate the SCHall phenomenon, we construct a curated benchmark, namely SCBench, comprising triplets: one or more simple visual-centric questions, a simple textual-centric question, and their corresponding compositional question. To ensure diversity, the visual questions cover object classification [16, 10], attribute recognition [12, 31], and OCR recognition [12, 48], while the textual ones span commonsense inference [41, 71], factual verification [46, 57], and numerical reasoning [44, 65]. Triplets are generated semi-automatically: simple visual and textual questions are automatically created per image using GPT-3.5 [51], verified to be correctly answered by most LVLMs, including LLaVA series [37], Qwen-VL series [2], MiniGPT-4 [78] and InternVL series [7], then paired into compositional questions and manually filtered and revised for both quality and difficulty. Notably, our benchmark differs fundamentally from existing LVLM hallucination benchmarks, such as POPE [33] (Figure 1a), which primarily target isolated visual recognition errors. In contrast, we focus on failures that emerge from composing questions that are individually simple and reliably answered. It is also distinct from knowledge-centric VQA benchmarks (*e.g.*, OK-VQA [46]), where the bottleneck lies in the textual subproblems, corresponding to limitations in external knowledge. More importantly, neither the data construction process nor the evaluation in OK-VQA considers compositionality. Unlike recent reasoning benchmarks [44, 65, 69] that emphasize multi-step reasoning chains, we instead target single-step compositions that unexpectedly induce hallucination.

Further, we validate that the SCHall phenomenon is widespread across a variety of LVLMs, rather than being confined to isolated cases, as evidenced by evaluations on both our compositional benchmark and general-purpose benchmarks. As shown in Figure 1(c), LVLMs such as the LLaVA series [37, 38], InternVL [7], and GPT-4o [50] exhibit substantial performance drops—accuracy on compositional questions decreases by nearly 20% compared to their near-perfect accuracy on the corresponding standalone visual- and textual-centric questions. When evaluating general-purpose benchmarks (*e.g.*, MMBench [41], MME [12], MMVet [71]), we decompose each question into visual- and textual-centric sub-questions (detailed in Sec 3.1). While visual recognition is a major source of hallucination, a notable pattern emerges: when the visual sub-question is answered correctly, hallucinations less frequently result from errors in the textual sub-question, which is also more likely answered correctly. Instead, they arise from the composition of sub-questions that are otherwise independently answerable (Figure 2). Interestingly, this phenomenon is more pronounced in stronger models such as QwenVL[2] and InternVL [7] compared to LLaVA-1.5 [37]. As their visual recognition improves, hallucinations from visual errors decrease, while those caused by composition become more prominent—further underscoring the importance of studying the SCHall phenomenon.

To probe the underlying causes of the SCHall phenomenon, we conduct a series of analyses and identify two preliminary factors that may contribute to it. *First, LVLMs struggle with composition-ality, particularly in targeting relevant visual content, more notably, abstracting it into textual understanding.* We find that masking irrelevant visual regions—forcing the model to rely solely

on relevant content—significantly improves performance, indicating that compositionality hinders accurate targeting of critical visual information. Similarly, inserting textual cues into the question that directly reference key visual regions also yields gains, suggesting that failures in abstracting visual content into textual understanding can induce hallucinations even in simple compositional settings (see Sec. 3.2 for details). ***Second, the mere presence of visual input—even a blank canvas devoid of meaningful content—can degrade the model's language processing performance.*** We observe that on the ScienceQA dataset [43], attaching a blank image to purely textual questions—*i.e.*, those answerable without visual input—leads to a noticeable drop in performance. Moreover, when comparing compositional and textual-centric question pairs, the answer logits for compositional questions are substantially lower (often by half), and correct answers tend to appear later in the output sequence. All these findings (detailed in Sec. 3.3) suggest that language processing is disrupted in compositional settings.

Based on the aforementioned definitions and findings, we propose a novel baseline, VLR-distillation, as the first attempt to address SCHall. To promote effective visual information extraction and representation, we introduce an innovative token type, referred to as the Vision Language Registers (VLRs), which serves as a bridge between the visual and linguistic modalities. Designed to represent the question-relevant image information while also engaging in textual understanding like text tokens, the VLRs fulfills the roles of both visual localizers and abstract semantic encoders, thereby reducing the model's functional gap between recognition and compositional tasks. Furthermore, *to counter the degradation of language processing capabilities caused by visual integration, we introduce a textually-enhanced VLR-distillation strategy.* Leveraging the inherent strength of language models in textual reasoning, we employ a text-represented visual branch as the teacher to guide the LVLM student, enabling it to preserve its language processing while effectively incorporating visual context.

To validate our findings and the effectiveness of the VLR-distillation, extensive experiments across various benchmarks demonstrate that the VLRs and distillation learning strategy not only yield significant improvements on our SCBench but also prove effective on different hallucination benchmarks and general VQA benchmarks. This further supports the validity, necessity, and generalizability of our proposed SCHall for LVLMs.

In summary, our contributions are as follows: (1) We identify a pervasive and fundamental SCHall phenomenon and introduce SCBench to systematically assess it, revealing significant limitations in LVLMs and paving the way for advancing hallucination research. (2) We conduct a thorough analysis of the challenges associated with this phenomenon, *i.e.*, attending to relevant visual content while preserving accurate and fluent language processing. (3) We propose VLR-distillation to mitigate hallucination, yielding substantial improvements not only on SCBench but also across diverse hallucination benchmarks and general VQA benchmarks.

## 2 Simple-Composed Benchmark Construction

This section describes the data generation process of our SCBench benchmark. Following a bottom-up strategy, we first construct atomic questions targeting visual recognition and textual understanding. These are then composed into simple-composed questions. Finally, we perform cross-validation to filter out those that are likely to induce SCHall. For further details of SCBench benchmark, please refer to the Appendix.

**Atomic Questions Generation.** We collect images from established datasets, including MM-Bench [41], MME [12] and SEEDBench [31], as well as from various online sources. In addition, we manually synthesize images containing texts, numbers, and geometric shapes to increase diversity. For each image, we generate captions using popular LVLMs including LLaVA series [37], Qwen-VL series [2], MiniGPT-4 [78] and InternVL series [7], identifying common content "easily" recognized by these models. Based on these captions, we use GPT-3.5 [51] to formulate corresponding recognition questions, which are subsequently filtered and refined through manual verification.

To construct textual cognition questions, we prompt GPT-3.5 using the recognized content as contextual input, encouraging it to generate questions from diverse linguistic perspectives. After manual verification, we evaluate the same LVLMs on these questions and retain those with high accuracy as "easy" instances.

**Simple-Composed Question Generation.** By replacing the text-represented visual content in the textual cognition questions with corresponding image inputs, we construct candidate simple-

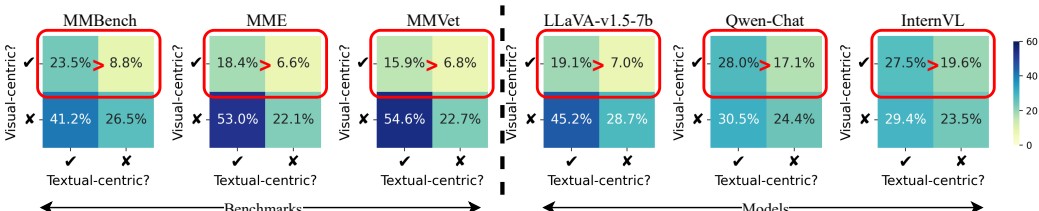

Figure 2: **Proportions of error attributed to visual recognition and textual understanding failures** across different benchmarks and models. When visual recognition is hallucination-free (the first line in each square), hallucinations occur more frequently in questions that have correctly answered text-centric sub-questions (top left corner) than in those with failed ones (top right corner).

composed questions, where both the visual recognition and textual components are individually "easy" for LVLMs. From these candidates, we identify questions that remain challenging for LVLMs—namely, those with relatively low accuracy across models—which are then manually reviewed and refined to construct the final benchmark.

# 3 Probing Simple-Composed Hallucinations

In this section, we first present statistical evidence that SCHall occurs in general-purpose benchmarks (see Sec. 3.1), supporting its broad prevalence, as also demonstrated by our benchmark introduced in Sec. 1. Based on our benchmark, we then examine two primary factors contributing to SCHall under compositional settings: (1) the model's failure to effectively target and abstract question-relevant visual content (see Sec. 3.2), and (2) the degradation of language processing pathways induced by the integration of visual inputs (see Sec. 3.3).

## 3.1 Beyond Isolated Flaws: Simple-Composed Hallucinations

**Motivation.** Research on hallucinations in LVLMs primarily emphasizes recognition errors. However, once these recognition tasks become "easy", do unique hallucinations specific to LVLMs continue to persist? In this context, we present empirical evidence (see Figure 1) supporting a negative conclusion. To further establish the universality of this hallucination, we conduct a statistical analysis over a diverse set of samples drawn from general-purpose benchmarks, evaluating multiple LVLM series. We include experiments with different decoding strategies in Appendix.

**Setting.** We uniformly sample a variety of question types from multiple datasets, including MME [12], MMBench [41], and MM-Vet [71]. Each question is annotated with its decomposed components, including one or more recognition and language questions that together cover the visual and linguistic capabilities required to answer the original question. Our experiments are conducted based on LLaVA1.5-7b [37], Qwen-VL [2] and InternVL [7], involving a sampled set of 300 instances.

**Result & Discussion.** The results are presented in Figure 2. As noted by previous research, recognition errors account for a substantial fraction of overall failures (41.2% & 26.5% on MMBench on the bottom-left corner). However, a considerable portion of the remaining errors associates with simple textual- and visual-centric questions. These errors are evident across both benchmarks and models dimensions, highlighting the prevalence of this hallucination: *Even when both the textual- and visual-centric questions are individually hallucination-free for LVLMs, their compositions can still pose unexpected challenges.* We further observe that the proportion of this type of hallucination increases from LLaVA (19.1%) to InternVL (27.5%), likely due to the latter's stronger visual recognition capability. This trend highlights the growing importance of addressing such hallucinations as LVLMs continue to improve in perceptual accuracy.

## 3.2 Visual Abstraction Fails under Compositional Questioning

**Motivation.** In recognition tasks, the model identifies relevant elements based on an explicit query. In contrast, compositional questions render the query implicit, as they often entail multiple intertwined sub-goals, thereby introducing new challenges. We thus hypothesize that one potential cause of

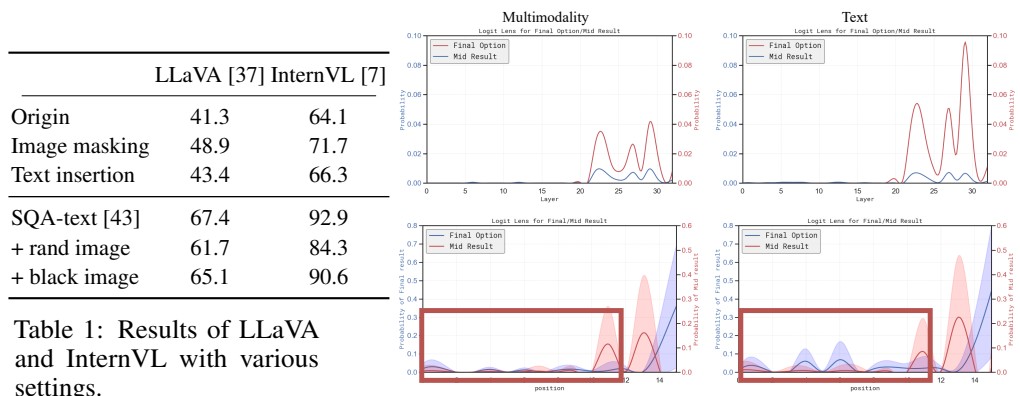

| | LLaVA [37] | InternVL [7] |
|---|---|---|
| Origin | 41.3 | 64.1 |
| Image masking | 48.9 | 71.7 |
| Text insertion | 43.4 | 66.3 |
| SQA-text [43] | 67.4 | 92.9 |
| + rand image | 61.7 | 84.3 |
| + black image | 65.1 | 90.6 |

Table 1: Results of LLaVA and InternVL with various settings.

Figure 3: **Analysis**: Logit Lens analysis on our benchmark.

SCHall may lies in **failures of visual abstraction** triggered by implicit queries in compositional settings.

**Setting.** To validate the hypothesis, we use two input modification strategies to provide recognition cues: (a) Image masking, where the original image is masked to retain only the region corresponding to the queried content; (b) Text insertion, where additional textual tokens are inserted to highlight the relevant visual content. The inserted text is constructed solely based on existing visual information in the question, ensuring no information leakage (see Appendix for details). We conduct experiments using LLaVA-1.5-7B and InternVL on a subset of our benchmark.

**Result & Discussion.** As shown in Table 1, both manipulations reduce SCHall, supporting our hypothesis: while the model attend effectively to relevant regions in isolated recognition tasks, this selective ability becomes a bottleneck in compositional tasks. Besides, image masking proves more effective than text insertion, as it directly eliminates misleading visual input, whereas text insertion only provides additional contextual cues for visual abstraction. Notably, using manually annotated masks resulted in modest improvement (8%), suggesting that other underexplored factors may contribute to SCHall.

### 3.3 Visual Inputs Induce Degradation in Language Processing

**Motivation.** However, what happens when the visual input is simple and free of distractions? We find that the model still exhibits the SCHall phenomenon. A typical failure mode involves the model conduct directly matching when answering questions, while neglecting the other context, exhibiting shortcut reasoning behavior. These observations lead us to hypothesize that the integration of visual information disrupts the language processing capability, ultimately giving rise to SCHall.

**Setting.** We conduct both statistical and visualization analyses on LLaVA1.5-7B to verify this hypothesis. (a) We first focus on 1,434 text-only questions from ScienceQA. To assess the influence of visual input, we pair each sample with an unrelated or visually uninformative black image, and examine the resulting changes in model performance. (b) To better understand the mechanism, we employ a logit lens analysis across transformer layers and token positions and take averages based on the benchmark. We focus on the final answer token, as well as intermediate result tokens that correspond to sub-answers of decomposed recognition questions. This approach enable us to trace how visual and linguistic signals are progressively integrated by the model during inference. Please refer to appendix for more details.

**Result & Discussion.** (a) Table 1 shows a notable 5.78% performance drop when paired with unrelated images, and a 2.1% degradation with black images, revealing that visual inputs can negatively affect the model's language processing capabilities—even when the visual input contains no meaningful content. (b) As shown in Figure 3, the layer-wise visualizations at the final input token (the first line) indicate that the answer logits in the purely textual condition are significantly higher than those in the compositional condition—often nearly twice as large. While the second line in Figure 3 reveal a positional distinction: multimodal inputs exhibit notably weaker activations at earlier token positions compared to text-only inputs. This finding suggest a delay in model's language processing under compositional settings, validating our hypothesis.

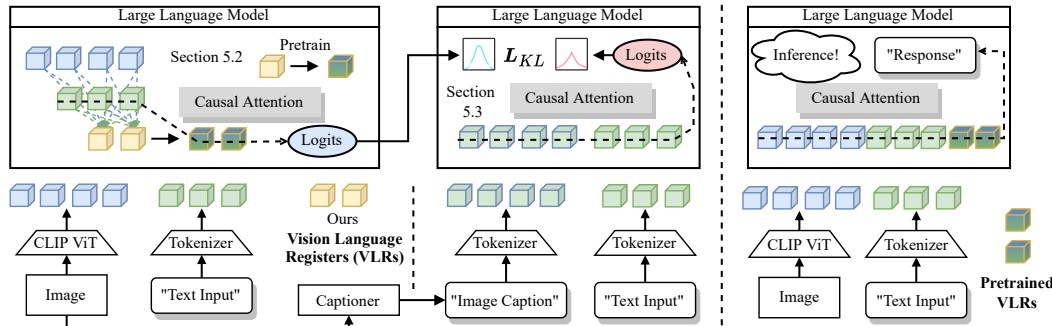

Figure 4: **An overview of our VLR-distillation.** Training Stage 1: Pretrain the VLRs with masked self-attention to learn selective image querying. Training Stage2: Distillation learning via a teacher branch augmented with additional captions, providing enhanced language-guided supervision. Inference: Use pretrained VLRs in the same manner as training to generate responses.

## 4 VLR-Distillation Method

An overview of our proposed method, VLR-Distillation, is depicted in Fig. 4. We begin with a preliminary of LVLMs in Sec. 4.1. Next, we introduce Vision Language Registers (VLRs) (Sec. 4.2) and employ a distillation learning strategy (Sec. 4.3) to alleviate the perturbations in language processing that arise when visual inputs are integrated.

### 4.1 Preliminary

**The Architecture of LVLMs.** We consider LVLMs as consisting of a vision encoder, a vision-language connector, and a language model. Given an input image $I$ and an instruction $T$, the vision encoder processes the image to extract features, which are then projected into a text-aligned feature space. Simultaneously, the instruction $T$ is tokenized into text tokens $\boldsymbol{X}_T$, with their embeddings computed for subsequent processing. These projected visual features $\boldsymbol{X}_I$ and text embeddings $\boldsymbol{X}_T$ serve as inputs to the language model. The reasoning process of the language model, leading to the output $\boldsymbol{Y}$, can be articulated as follows:

$$p(\boldsymbol{Y}|\boldsymbol{X}_I, \boldsymbol{X}_T) = \prod_{i\in\mathcal{L}} p_\theta(y_i|[\boldsymbol{X}_I, \boldsymbol{X}_T, \boldsymbol{Y}_{<i}]) \tag{1}$$

where $\mathcal{L}$ denotes the set of output positions, and $\boldsymbol{Y}_{<i}$ is predicted output before current token $y_i$.

**Causal Attention.** Following LLMs, LVLMs employ causal attention to ensure that each position is unable to access information from future positions:

$$\boldsymbol{M}_{r,s} = \begin{cases} True, r < s \\ False, \text{otherwise} \end{cases} \tag{2}$$

where $\boldsymbol{M}_{r,s}$ indicates whether the token at position $r$ has access to the token at position $s$.

### 4.2 Vision Language Registers: Absorbing Visual Content conditioned on Language

To promote effective visual information extraction and, we introduce additional VLRs that selectively absorb relevant visual content aligned with the input textual query. Specifically, we randomly initialize $N$ tokens in the feature space of text tokens, denoted as the sequence $\boldsymbol{X}_{VLRs}$. This sequence is then concatenated with both the image input and the instruction input, enabling the generation of the answer $\boldsymbol{Y}$ to be expressed as follows:

$$p(\boldsymbol{Y}|\boldsymbol{X}_I, \boldsymbol{X}_T) = \prod_{i\in\mathcal{L}} p_\theta(y_i|[\boldsymbol{X}_I, \boldsymbol{X}_T, \boldsymbol{X}_{VLRs}, \boldsymbol{Y}_{<i}]). \tag{3}$$

**Pretraining.** To ensure that the randomly initialized VLRs exhibits visual abstraction capability, we first pretrain VLRs in which all other components of LVLMs remain fixed. We modify the attention

mask to prevent output tokens from directly attending to image tokens, thereby compelling the VLRs to serve as a bridge by effectively aggregating information from the image tokens, as follows:

$$\boldsymbol{M}'_{r,s} = \begin{cases} False, r \in \mathcal{L} \text{ and } s \in \boldsymbol{X}_I \\ \boldsymbol{M}_{r,s}, \text{otherwise} \end{cases} \tag{4}$$

### 4.3 Distillation Learning from Textual Enhanced Branch

To mitigate the degradation of language processing capabilities under compositional conditions, we propose a Distillation Learning strategy. Specifically, we leverage the strength of language models on text-only questions to guide the model in preserving its inherent linguistic competence during multimodal inference. The text-only branch substitutes the image input in the form of image tokens $\boldsymbol{X}_I$ with image captions tokenized as text tokens $\boldsymbol{X}_c$. The approach can be expressed as follows:

$$p(\boldsymbol{Y}'|\boldsymbol{X}_c, \boldsymbol{X}_T) = \prod_{i \in \mathcal{L}} p_\theta(y'_i|[\boldsymbol{X}_c, \boldsymbol{X}_T, \boldsymbol{Y}'_{<i}]) \tag{5}$$

We subsequently compute the Kullback-Leibler divergence between the introduced text-only branch (5) and the original branch (3), denoted as:

$$\boldsymbol{L}_{KL} = E_{V,T}[D_{KL}(p(\boldsymbol{Y}'|\boldsymbol{X}_c, \boldsymbol{X}_T)||p(\boldsymbol{Y}|\boldsymbol{X}_I, \boldsymbol{X}_T))]. \tag{6}$$

The final loss $\boldsymbol{L}$ is formulated as follows:

$$\boldsymbol{L} = \boldsymbol{L}_{reg} + \boldsymbol{L}'_{reg} + \boldsymbol{L}_{KL} \tag{7}$$

where $\boldsymbol{L}_{reg}$ and $\boldsymbol{L}'_{reg}$ indicates the language modeling loss of the original branch and the text-only branch.

## 5 Experiments

### 5.1 Experimental Settings

**Datasets and Baselines.** To evaluate the effectiveness of our method across various architectures, we experiment with LLaVA1.5 [37], Qwen-VL-Chat [2], and MiniGPT-4 [78] as primary baselines. *For training*, we employ a subset of the training data from the instruction tuning (IT) phase of these models. Given that MiniGPT-4 is trained exclusively on caption data, it exhibits limited capability in addressing broader VQA tasks. Therefore, we finetune MiniGPT-4 using a subset of the IT training data from LLaVA1.5 as the baseline, which also serves as the training data for our proposed method. *For inference*, we first conduct experiments on our proposed SCBench, comparing our methods with popular hallucination mitigating methods, to demonstrate the effectiveness of our proposed VLR-distillation. Additionally, we report results on popular hallucination benchmarks including POPE [33], MME-hall [12] and general-pupose VQA benchmarks encompassing ScienceQA [43], MMBench [41], HallusionBench [14] and MM-Vet [71].

**Implementation details.** *For training*, we have two training phases: pretraining stage for VLRs and distillation learning, both following the alignment learning and instruction tuning stages of the baseline model. During pretraining, we use a batch size of 128, freezing all other parts of the model and training only the VLRs. In the distillation learning phase, we employ a batch size of 64 with 2 accumulation steps, freezing the pretrained VLRs and training the LoRA [19] of the language model. For each baseline, we set the number of VLRs $N$ to 4. All experiments are conducted for a single epoch, utilizing the Adam optimizer on 8 A100 GPUs. *For inference*, we follow VCD [29] using nucleus sampling for experiments on POPE and MME, while applying greedy decoding for other benchmarks.

### 5.2 Comparison on Simple-Composed Benchmark

Table 2 presents the results of standard baselines, representative hallucination mitigation methods, and our proposed VLR-distillation approach on the SCBench benchmark. Despite the low complexity of the questions in our SCBench benchmark, all three baselines perform poorly, achieving an average

| Model | Score in Various Question Type | | | | | | | |
|---|---|---|---|---|---|---|---|---|
| | Perc. | Sci. | Comm. | Fact | Lang. | Scene | Math | **Overall** |
| LLaVA1.5-7b | 46.55 | 30.61 | 43.59 | 46.67 | 36.67 | 23.33 | 19.30 | 33.75 |
| + VCD[29] | 48.28 | 36.73 | 41.03 | 46.67 | 36.67 | 26.67 | 21.05 | 35.60 |
| + PAI[40] | 32.76 | 24.49 | 28.21 | 16.67 | 40.00 | 13.33 | 14.04 | 23.22 |
| + CODE[28] | 46.55 | 32.65 | 46.15 | 50.00 | 40.00 | 25.00 | 21.05 | 35.60 |
| + REVERIE[72] | 48.28 | 32.65 | 43.59 | 40.00 | 43.33 | 21.67 | 26.32 | 35.29 |
| + CCA[67] | 44.83 | 38.78 | 35.90 | 30.00 | 36.67 | 28.33 | 22.81 | 33.75 |
| + ours | **51.72** | **46.94** | **48.72** | **53.33** | **46.67** | **28.33** | **28.07** | **41.80** |
| | +5.17 | +16.33 | +5.13 | +6.66 | +10.00 | +5.00 | +8.77 | +8.05 |
| QwenVL-Chat | 51.72 | 42.86 | 53.85 | 63.33 | 23.33 | 41.67 | 24.56 | 42.41 |
| + VCD[29] | 51.72 | 42.86 | 53.85 | 66.67 | 23.33 | 38.33 | 28.07 | 42.72 |
| + PAI[40] | 56.90 | 42.86 | 43.59 | 50.00 | 20.00 | 31.67 | 24.56 | 38.70 |
| + CODE[28] | 50.00 | 46.94 | 53.85 | 60.00 | 23.33 | 38.33 | 22.81 | 41.49 |
| + ours | **56.90** | **48.98** | **58.97** | **70.00** | **26.67** | **43.33** | **29.82** | **47.06** |
| | +5.19 | +6.12 | +5.12 | +6.67 | +3.34 | +1.66 | +5.26 | +4.65 |
| MiniGPT-4 | 22.41 | 12.24 | 17.95 | 6.67 | 23.33 | 16.67 | 5.26 | 14.86 |
| + VCD[29] | 24.14 | 14.29 | 20.51 | 6.67 | 23.33 | 16.67 | 10.53 | 16.72 |
| + PAI[40] | 18.97 | 24.49 | 15.38 | 23.33 | 36.67 | 18.33 | 1.75 | 18.27 |
| + CODE[28] | 15.52 | 16.33 | 17.95 | 10.00 | 26.67 | 15.00 | 10.53 | 15.48 |
| + ours | **32.76** | **34.69** | **25.64** | **30.00** | **30.00** | **18.33** | **21.05** | **26.93** |
| | +10.35 | +22.45 | +7.69 | +23.33 | +6.67 | +1.66 | +15.79 | +12.07 |

Table 2: **Results on Our SCBench Benchmark.** Although their decomposed visual-centric and texual-centric questions are hallucination-free, LVLMs struggle with this "simple" dataset. Full names of the categories in our benchmark: *Perception*, *Science*, *Commonsense Reasoning*, *Factual Knowledge*, *Language Capability*, *Scene Understanding* and *Math*.

accuracy of approximately 30%. These results are consistent with our analysis in Sec. 3, which suggests that LVLMs are prone to ScHall.

We also evaluate several popular hallucination mitigation strategies, including zero-shot methods, including VCD [29], PAI [40] and CODE [28], as well as training approaches, including REVERIE [72] and CCA [67]. The experimental results indicate that while these methods can suppress object hallucinations, they do not perform well in mitigating the ScHall. Compared to existing methods that focus solely on object hallucinations, our approach consistently achieves substantial improvements on LLaVA1.5-7B, Qwen-VL, and MiniGPT-4 on our SCBench benchmark. In particular, it yields notable gains of 8.05% and 12.07% in overall accuracy on LLaVA1.5-7B and MiniGPT-4, respectively, indicating its effectiveness in activating the models' latent capabilities under composed scenarios.

| Setting | Model | w/ ours | Accuracy↑ | Precision | Recall | F1 Score↑ |
|---|---|---|---|---|---|---|
| *Random* | LLaVA1.5 | ✗ | $83.29_{(\pm0.35)}$ | $92.13_{(\pm0.54)}$ | $72.80_{(\pm0.57)}$ | $81.33_{(\pm0.41)}$ |
| | | ✓ | $\mathbf{87.46}_{(\pm0.42)}$ | $92.04_{(\pm0.49)}$ | $82.06_{(\pm0.77)}$ | $\mathbf{86.76}_{(\pm0.49)}$ |
| | Qwen-VL | ✗ | $84.73_{(\pm0.36)}$ | $95.61_{(\pm0.45)}$ | $72.81_{(\pm0.38)}$ | $82.67_{(\pm0.41)}$ |
| | | ✓ | $\mathbf{87.59}_{(\pm0.44)}$ | $93.68_{(\pm0.69)}$ | $80.63_{(\pm0.47)}$ | $\mathbf{86.66}_{(\pm0.47)}$ |
| | MiniGPT-4 | ✗ | $74.85_{(\pm0.27)}$ | $80.50_{(\pm0.82)}$ | $65.60_{(\pm0.52)}$ | $72.28_{(\pm0.19)}$ |
| | | ✓ | $\mathbf{83.99}_{(\pm0.35)}$ | $90.78_{(\pm0.62)}$ | $75.68_{(\pm0.80)}$ | $\mathbf{82.54}_{(\pm0.44)}$ |
| *Popular* | LLaVA1.5 | ✗ | $81.88_{(\pm0.48)}$ | $88.93_{(\pm0.60)}$ | $72.80_{(\pm0.57)}$ | $80.06_{(\pm0.05)}$ |
| | | ✓ | $\mathbf{85.28}_{(\pm0.17)}$ | $87.02_{(\pm0.39)}$ | $83.02_{(\pm0.52)}$ | $\mathbf{84.98}_{(\pm0.19)}$ |
| | Qwen-VL | ✗ | $84.13_{(\pm0.18)}$ | $94.31_{(\pm0.43)}$ | $72.64_{(\pm0.45)}$ | $82.06_{(\pm0.23)}$ |
| | | ✓ | $\mathbf{85.68}_{(\pm0.22)}$ | $89.88_{(\pm0.23)}$ | $80.41_{(\pm0.32)}$ | $\mathbf{84.88}_{(\pm0.25)}$ |
| | MiniGPT-4 | ✗ | $71.85_{(\pm0.64)}$ | $74.70_{(\pm0.69)}$ | $66.09_{(\pm0.90)}$ | $70.13_{(\pm0.74)}$ |
| | | ✓ | $\mathbf{80.45}_{(\pm0.23)}$ | $84.04_{(\pm0.68)}$ | $75.20_{(\pm0.77)}$ | $\mathbf{79.37}_{(\pm0.27)}$ |
| *Adversarial* | LLaVA1.5 | ✗ | $78.96_{(\pm0.52)}$ | $83.06_{(\pm0.58)}$ | $72.75_{(\pm0.59)}$ | $77.57_{(\pm0.57)}$ |
| | | ✓ | $\mathbf{81.18}_{(\pm0.41)}$ | $80.24_{(\pm0.67)}$ | $82.96_{(\pm0.32)}$ | $\mathbf{81.56}_{(\pm0.41)}$ |
| | Qwen-VL | ✗ | $82.26_{(\pm0.30)}$ | $89.97_{(\pm0.33)}$ | $72.61_{(\pm0.50)}$ | $80.37_{(\pm0.37)}$ |
| | | ✓ | $\mathbf{82.86}_{(\pm0.27)}$ | $84.37_{(\pm0.33)}$ | $80.67_{(\pm0.30)}$ | $\mathbf{80.48}_{(\pm0.28)}$ |
| | MiniGPT-4 | ✗ | $70.19_{(\pm0.43)}$ | $72.03_{(\pm0.59)}$ | $66.03_{(\pm0.59)}$ | $68.90_{(\pm0.44)}$ |
| | | ✓ | $\mathbf{78.13}_{(\pm0.10)}$ | $79.70_{(\pm0.30)}$ | $75.49_{(\pm0.44)}$ | $\mathbf{77.54}_{(\pm0.14)}$ |

Table 3: **Results on POPE MSCOCO.** The best performances for baselines is highlighted in **bolded**.

| Model | w/ ours | Object-level | | Attribute-level | | Total Scores↑ |
|---|---|---|---|---|---|---|
| | | *Existence↑* | *Count↑* | *Position↑* | *Color↑* | |
| LLaVA1.5 | ✗ | $181.00_{(\pm 5.83)}$ | $96.67_{(\pm 7.89)}$ | $105.00_{(\pm 11.69)}$ | $127.67_{(\pm 15.55)}$ | $510.33_{(\pm 26.65)}$ |
| | ✓ | $\mathbf{191.00}_{(\pm 3.74)}$ | $\mathbf{135.67}_{(\pm 6.38)}$ | $\mathbf{116.33}_{(\pm 15.22)}$ | $\mathbf{142.67}_{(\pm 11.48)}$ | $\mathbf{585.67}_{(\pm 22.89)}$ |
| Qwen-VL | ✗ | $180.83_{(\pm 5.34)}$ | $120.83_{(\pm 10.13)}$ | $115.28_{(\pm 2.24)}$ | $168.61_{(\pm 8.36)}$ | $585.56_{(\pm 12.46)}$ |
| | ✓ | $\mathbf{186.67}_{(\pm 2.36)}$ | $\mathbf{134.44}_{(\pm 12.27)}$ | $\mathbf{123.89}_{(\pm 6.43)}$ | $\mathbf{173.33}_{(\pm 8.55)}$ | $\mathbf{618.33}_{(\pm 14.81)}$ |
| MiniGPT-4 | ✗ | $142.33_{(\pm 7.02)}$ | $69.00_{(\pm 10.20)}$ | $63.33_{(\pm 13.46)}$ | $97.33_{(\pm 15.83)}$ | $371.67_{(\pm 25.25)}$ |
| | ✓ | $\mathbf{168.67}_{(\pm 4.14)}$ | $\mathbf{88.33}_{(\pm 11.79)}$ | $\mathbf{71.67}_{(\pm 7.75)}$ | $\mathbf{111.67}_{(\pm 4.83)}$ | $\mathbf{440.33}_{(\pm 7.10)}$ |

Table 4: **Results on MME-hall.** Higher scores indicate better performance and fewer hallucinations.

## 5.3 Comparisons on Other Hallucination Benchmarks and General-purpose Benchmarks

**POPE.** The results on the POPE dataset are detailed in Table 3. Our method achieves substantial improvements across the random, popular, and adversarial setups for LLaVA1.5, Qwen-VL, and MiniGPT-4. Notably, we observe enhancements on accuracy of $+9.14$, $+8.60$, and $+7.94$ over the MiniGPT-4 baseline in the three respective setups. Furthermore, our method shows a significant improvement in recall, with average values of $+9.05$, $+8.97$, and $+9.12$ in the three setups, highlighting that our VLRs prioritize visual information that is often overlooked and susceptible to interference from redundant data.

**MME.** As shwon in Table 4, our method performs favorably on the benchmark, showing consistent improvements over baseline models across all splits. Notably, we achieve an improvement of 75 on LLaVA baseline.

**General-purpose Benchmarks**. As shown in Table 5, we experiment on MMBench [41] for comprehensive evaluation, ScienceQA [43] for scientific questions, HallusionBench [14] for challenging hallucination questions, and MM-Vet [71] for open-ended hallucination questions. Our method consistently demonstrates an improvement of approximately 1.5 across these general benchmarks. Notable advancements, $+1.9$ and $+2.2$, are demonstrated in the HallusionBench and MM-Vet, which focus on hallucinations.

| Model | MMB [41] | SQA [43] | Hallusion Bench [14] | MM-Vet [71] |
|---|---|---|---|---|
| BLIP-2 [32] | - | 61.0 | - | 22.4 |
| InstructBLIP [9] | 39.8 | 63.1 | 45.26 | 25.6 |
| MiniGPT-4 [78] | 30.5 | - | 35.78 | 22.1 |
| Qwen-VL [2] | 38.2 | 67.1 | 39.15 | - |
| Qwen-VL-Chat [2] | 60.6 | 68.2 | - | - |
| LLaVAv1.5 [37] | 64.3 | 66.8 | 47.6 | 31.1 |
| + ours | 65.4 | 67.8 | 49.5 | 33.3 |

Table 5: **Results on the general-purpose VQA benchmarks.**

## 5.4 Ablation Study

| VLRs | DL | Random | | Popular | | Adversarial | |
|---|---|---|---|---|---|---|---|
| | | Acc | F1 | Acc | F1 | Acc | F1 |
| ✗ | ✗ | 83.3 | 81.3 | 81.9 | 80.1 | 79.0 | 77.6 |
| ✓ | ✗ | 86.5 | 85.9 | 84.1 | 83.7 | 80.2 | 80.4 |
| ✗ | ✓ | 85.0 | 83.4 | 84.3 | 83.0 | 81.4 | 80.4 |
| ✓ | ✓ | 87.5 | 86.8 | 85.3 | 85.0 | 81.2 | 81.6 |

| #VLRs | Random | | Popular | | Adversarial | |
|---|---|---|---|---|---|---|
| | Acc | F1 | Acc | F1 | Acc | F1 |
| 2 | 87.4 | 87.1 | 84.7 | 84.8 | 80.2 | 81.1 |
| 4 | 87.5 | 86.8 | 85.3 | 85.0 | 81.2 | 81.6 |
| 8 | 87.7 | 87.4 | 85.3 | 85.2 | 80.1 | 80.8 |
| 16 | 87.0 | 86.3 | 84.7 | 84.3 | 80.9 | 81.5 |

Table 6: **Ablation study** on POPE using LLaVA-v1.5 baseline.

To demonstrate the effectiveness of our proposed VLRs and distillation learning training strategies, we conduct ablation studies on POPE based on LLaVA1.5 baseline, as shown in Table 6. (1) It can be observed that each component—VLRs and the distillation learning strategy—individually contributes to an improvement in the model's performance on POPE. (2) It is noteworthy that the

independently used VLRs result in a surprisingly significant improvement. This indicates that the VLRs, as a supplementary visual component, aids the model in recognition. (3) Additionally, the distillation learning strategy for the caption branch, which reduce the internal gap on language-side, allows the model to learn to moderately prioritize image information and perform question-answering based on the primary visual content.

# 6 Related Work

**Hallucinations in LLMs.** The generation of meaningless or unfaithful outputs—commonly referred to as *hallucinations* [47, 54, 55]—in natural language generation has garnered considerable attention, as it poses significant risks to real-world applications of language models, particularly large language models (LLMs) [62, 8, 1]. In the context of LLMs, hallucinations are typically classified into two primary types: factual hallucinations [47, 60, 21], where the generated output contradicts or cannot be verified by real-world facts, and fidelity hallucinations [21], where the output deviates from the input or fails to remain consistent with preceding output.

**Hallucinations in LVLMs.** Unlike hallucinations observed in LLMs, LVLMs introduce *object hallucinations*, where generated content misaligns with the visual input [55, 33]. This issue is commonly attributed to language priors [66, 36], statistical bias [66, 77], or modality gap [26]. Existing efforts mitigate object hallucinations by improving model architectures [61, 7, 67], curating training datasets [36, 77, 70, 72], designing learning strategies [26], or leveraging the intrinsic properties of pre-trained LVLMs [66, 22, 40, 28, 45].

Additionally, several studies discuss other types of hallucinations in LVLMs beyond object hallucinations. LRV [36] observe instruction-following failures, while more recent studies emphasize the issue of new types of visual hallucinations, including multi-object hallucinations [6], event hallucination [25] and prompted visual hallucination [39], respectively. Some recent approaches focus on probing the internal mechanisms [58] of LVLMs to attribute hallucinations, such as identifying the different causal pathways that lead to hallucinations [53] or understanding why longer contexts are more prone to causing them [75].

**Benchmarks for Hallucination in LVLMs.** To facilitate the study of hallucination in LVLMs, several benchmarks have been proposed, most of which primarily focus on object hallucinations. Early efforts, such as CHAIR, [55], concentrates on hallucinated objects in image captioning. Subsequent benchmarks [33, 12, 14, 42] adopt more structured formats, including yes/no and multiple-choice questions settings, to simplify evaluation. More recent efforts expand both the scope and evaluation protocols. For generative tasks, GPT-based tools [36, 59, 71] offer flexible, context-aware evaluation, while FaithScore [27] provides fine-grained faithfulness assessment. On the discriminative side, recent benchmarks [20, 25, 39] go beyond objects to include attributes and relational inconsistencies.

# 7 Conclusion

This paper focuses on a phenomenon in LVLMs: despite accurately answer questions in isolated textual- and visual-centric questions, it still struggles in the compositional one. We also establish a benchmark and conduct analysis. We further propose VLR-distillation and achieve high performance on our benchmarks and published ones. **Limitation**: It is important to acknowledge the potential ethical implications arising from LVLMs. Since our method leverages large vision language models like Llava and GPT-4o, it may also inherit biases and limitations present in these models.

**Acknowledgment:** This work is supported by the National Natural Science Foundation of China under Grant No.62206174 and No.62576365.

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

# A  Appendix Overview

This appendix provides further details on the SCBench benchmark, experimental settings, and additional results to support the main paper. The contents are organized as follows:

- Sec B - Details of SCBench Benchmark
    - Sec B.1 - Data Distributions
    - Sec B.2 - Data Sources
    - Sec B.3 - Details and Prompts for Data Construction
- Sec C - Analysis Details and Supplementary Results
    - Sec C.1 - Analysis on various decoding strategies
    - Sec C.2 - Detailed settings for image masking and text insertion experiments
    - Sec C.3 - Detailed settings for logit lens analysis
- Sec D - Additional Experimental Settings and Results
    - Sec D.1 - Additional Implementation Details
    - Sec D.2 - Additional Experiments on POPE
    - Sec D.3 - Additional Experiments on MME Remaining Subset
- Sec E - Visualizations

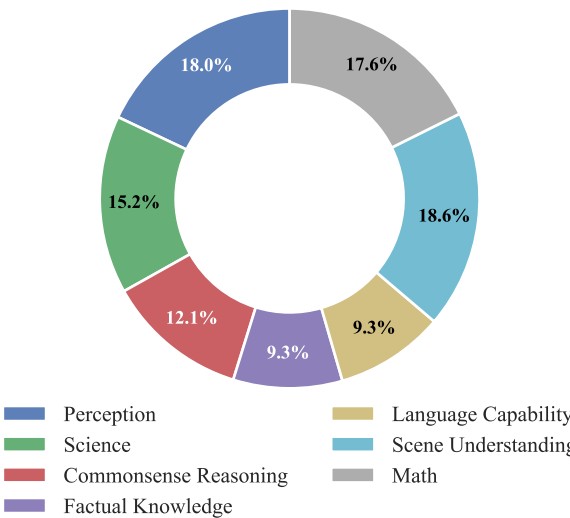

Figure 5: **Data distribution of SCBench benchmark.**

| Image Sources | Number | Proportion |
|---|---|---|
| COCO [35] | 20 | 6.2% |
| MMBench [41] | 98 | 30.3% |
| MME [12] | 36 | 11.1% |
| ScienceQA [43] | 15 | 4.6% |
| Internet | 124 | 38.4% |
| Constructed | 30 | 9.3% |

| Problem Sources | Number | Proportion |
|---|---|---|
| ScienceQA [43] | 15 | 4.6% |
| WinoGrande [56] | 7 | 2.2% |
| MMLU [17] | 10 | 3.1% |
| WSC [30] | 6 | 1.9% |
| StoryCloze [49] | 5 | 1.5% |
| MNLI [63] | 7 | 2.2% |
| QQP [63] | 5 | 1.5% |
| GPT-3.5 Generated | 268 | 83.0% |

Table 7: Data sources of SCBench Benchmark.

# B  Details of SCBench Benchmark

This section presents a detailed overview of the SCBench benchmark, covering the dataset distribution, the specific data sources utilized, and the prompt design strategy employed during dataset construction. Visualizations of representative examples are provided in Appendix E.

## B.1  Data Distributions

We construct the SCBench benchmark, comprising 951 questions in total—323 compositional and 628 decomposed—curated from diverse perspectives. The distribution of question types is visualized in Figure 5.

The dataset primarily focuses on questions that involve both visual- and textual-centric decomposed questions, accounting for 82% of the total. To address a distinct class of failures, we introduce

| GPT-3.5 Prompt |
| --- |
| Given a fact about an image, transform this fact into a concise and relevant question, and provide a corresponding answer. The question must explicitly include the word "image" and be appropriate to the level of the fact (object, attribute, relation, or event).

Format your response strictly as follows:
Question: [Your generated question]
Answer: [Your generated answer]

Below are several examples:

Object-level example:
Input Fact: There is a dog in the image.
Question: What is the animal in the image?
Answer: Dog.

Attribute-level example:
Input Fact: There is a white dog in the image.
Question: What is the color of the animal in the image?
Answer: White.

Relation-level example:
Input Fact: The dog is lying on the bench.
Question: What is the relation between the dog and the bench?
Answer: The dog is lying on the bench.

Event-level example:
Input Fact: The dog is sleeping.
Question: What is the dog doing?
Answer: The dog is sleeping.

Now, apply this format to the following input:
Input Fact: {fact}. |

Table 8: Prompts used for visual-centric atomic question generation in the SCBench construction pipeline.

a *Perception* category. This category captures cases where the model correctly identifies relevant content but still fails to answer accurately when the information is reformulated in MCQ format. These failures represent a specific type of compositional challenge, in which additional textual choices hinder accurate comprehension. By including these examples in the perception split, we aim to improve the overall coverage of the benchmark. Additionally, we introduce a *Language Capability* category, specifically designed to evaluate models' abilities to handle complex linguistic phenomena.

## B.2 Data Sources

We provide the sources of the images and questions included in our benchmark, as detailed in the Table 7. Most questions are carefully constructed following the pipeline described in the main text. Only language capability questions and a portion of science questions are adapted from existing NLP datasets [56, 17, 30, 49, 63] and the ScienceQA [43] dataset, respectively.

## B.3 Details and Prompts for Data Construction

**Visual-centric atomic question construction.** As introduced in the main text, we first prompt popular LVLMs with diverse captioning instructions to identify commonly recognized content—such as objects, attributes, relations, and events. Based on this content, we then construct visual-centric questions using the prompt template shown in Table 8.

**Textual-centric atomic question construction.** Based on commonly recognized image content, we prompt GPT-3.5 to generate questions and options. For each category, we first obtain a set of diverse perspectives using GPT-3.5 (e.g., typical animal behavior in *Science*) and formulate questions

| GPT-3.5 Prompt |
| --- |

You are an imaginative and highly creative language model. Given the caption of an image and a question related to this image, your task is to generate five correct answers and five incorrect answers for the given question.

Your answers should be realistic, logically sound, and plausible. Correct answers must accurately address the question, while incorrect answers should be clearly wrong or misleading, yet still sound superficially plausible. The answers do not have to be grounded in the image caption, but may optionally relate to it.

Strictly follow the format below:

Example:
Image caption: The image shows a dog lying on a bench at sunset.
Question: Which of the following is not typically a behavior exhibited by the animal in this image?
Correct answers:
1. Lying on a bench
2. Being very lazy
3. Writing with a pen
4. Using a litter box
5. Climbing trees
Incorrect answers:
1. Barking at strangers
2. Wagging their tails
3. Digging holes
4. Sniffing around
5. Herding sheep

Now, apply this format to the following input:
Image caption: {image caption}
Question: {question}

Table 9: Prompts used to generate answer options for textual-centric atomic questions in the SCBench construction pipeline. The input questions are also generated using GPT-3.5 with simple prompts to provide diverse perspectives on the given visual content across different categories (e.g., typical animal behavior in Science).

grounded in appropriate visual contexts (e.g., *Which of the following is not typically a behavior exhibited by the dog?*). We then prompt GPT-3.5 to generate corresponding answer options, using the template shown in Table 9.

**Exception on specific splits.** For the language capability split, we select questions from NLP datasets whose answers can be visually represented. Then we use concatenated images as image input, expressing answers through spatial references (e.g., "the image on the left/right" or "above/below"). For the science split, we adapt ScienceQA questions that are originally solvable without images into versions that require visual information for correct reasoning.

## C Analysis Details and Supplementary Results

### C.1 Analysis on various decoding strategies

As discussed in the main text, SCHall hallucinations are observed across a range of benchmarks and models. Here, we present additional experiments across different decoding strategies. We conduct experiments using the LLaVAv1.5-7b [37] model and the results are shown in Figure 6. The results demonstrate that SCHall is observed consistently across all decoding strategies.

### C.2 Detailed settings for image masking and text insertion experiments

**Image masking.** We use manually annotated images with masks. Specifically, we annotate bounding boxes that enclose the content necessary to answer the question, and mask out all other regions. An example is shown in the Figure 7 (a).

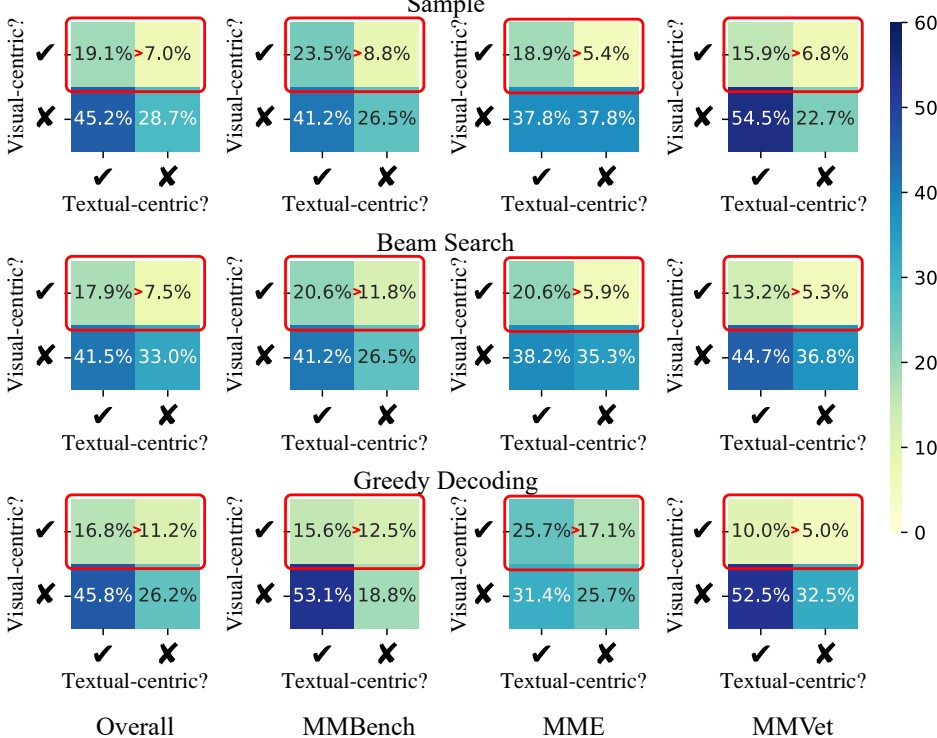

Figure 6: **Proportions of error attributed to recognition and textual understanding failures** across different decoding settings. When visual recognition is hallucination-free (the first line in each square), hallucinations occur more frequently in questions that have correctly answered text-centric sub-questions (top left corner) than in those with failed ones (top right corner). This phenomenon occurs consistently across sampling, beam search, and greedy decoding strategies on all datasets.

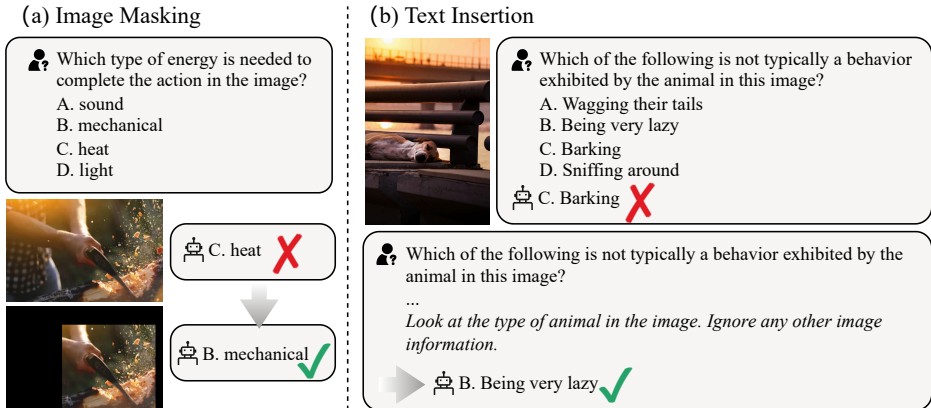

Figure 7: Examples for image masking and text insertion experiments with LLaVAv1.5.

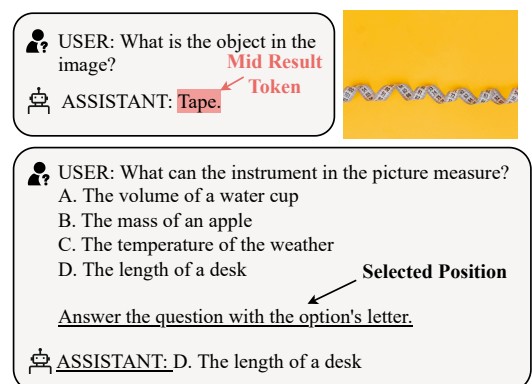

Figure 8: An example of logit lens analysis. The red-highlighted "tape" indicates the intermediate token we trace. Underlined tokens mark the positions of interest that we focus on and visualize.

**Text insertion.** We prepend a simple textual prompt, "Look at {image content}. Ignore any other image information.", to explicitly highlight the relevant visual content. The placeholder {image content} is populated with text extracted from the question itself, as illustrated in Figure 7 (b).

### C.3 Detailed settings for logit lens analysis

**Target tokens.** We visualize two types of tokens: the final answer token and the intermediate result token. The final answer token corresponds to the ground-truth answer (e.g., if the answer is D, we track the probability of token D in the logit lens). The intermediate result token refers to the token associated with the answer to the decomposed visual-centric sub-question. For example, in the question shown in Figure 8, the intermediate result token is the first token of "Tape". This setup allows us to examine the model's reasoning trajectory, where the intermediate result token is expected to appear earlier than the final answer token.

**Layer dimension.** To investigate how the probability of a target token evolves across layers, we compute its average probability over all input positions following the question prompt. For instance, in the example shown in Figure 8, the selected range spans from "Answer" to "ASSISTANT:" This yields a layer-wise trajectory of the target token's likelihood.

**Position dimension.** To analyze how the target token's probability changes across positions, we average its probability across all layers at each position. The visualized range also spans from "Answer" to "ASSISTANT:" resulting in a position-wise trajectory of the token's likelihood.

| Settings | LLaVAv1.5-7b | Qwen-VL-Chat | MiniGPT-4 |
|---|---|---|---|
| $a_1, a_2, a_3$ | | 1, 1, 1e5 | |
| batch size | | 128 | |
| lr | 2e-4 | 1e-5 | 3e-5 |
| lr schedule | | Cosine Decay | |
| lr warmup ratio | 0.03 | 0.01 | 0.05 |
| weight decay | 0 | 0.05 | 0.05 |
| epoch | | 1 | |
| optimizer | | AdamW | |
| DeepSpeed stage | 3 | 3 | / |

Table 10: Hyperparameters for our VLR-distillation methods. $a_1, a_2$ and $a_3$ are the coefficients for $L_{reg}, L'_{reg}$ and $L_{KL}$, respectively.

## D  Detailed Experimental Settings and Results

### D.1  Additional Implementation Details

In this section, we present the model-specific implementation details. For LLaVAv1.5-7b [37], we utilize a subset of its instruction-tuning datasets, specifically VQAv2 [13], OK-VQA [46], GQA [23],

| Dataset | Setting | Model | w/ ours | Accuracy↑ | Precision | Recall | F1 Score↑ |
|---|---|---|---|---|---|---|---|
| A-OKVQA | Random | LLaVA1.5 | ✗ | $83.45_{(\pm0.48)}$ | $87.24_{(\pm0.68)}$ | $78.36_{(\pm0.54)}$ | $82.56_{(\pm0.50)}$ |
| | | | ✓ | $\mathbf{87.57}_{(\pm0.37)}$ | $85.86_{(\pm0.44)}$ | $89.75_{(\pm0.50)}$ | $\mathbf{87.76}_{(\pm0.37)}$ |
| | | Qwen-VL | ✗ | $86.67_{(\pm0.48)}$ | $93.16_{(\pm0.55)}$ | $79.16_{(\pm0.59)}$ | $85.59_{(\pm0.53)}$ |
| | | | ✓ | $\mathbf{88.07}_{(\pm0.32)}$ | $89.13_{(\pm0.44)}$ | $86.72_{(\pm0.55)}$ | $\mathbf{87.91}_{(\pm0.34)}$ |
| | | MiniGPT-4 | ✗ | $72.38_{(\pm0.77)}$ | $75.66_{(\pm0.91)}$ | $66.00_{(\pm1.40)}$ | $70.49_{(\pm0.95)}$ |
| | | | ✓ | $\mathbf{80.08}_{(\pm0.68)}$ | $82.82_{(\pm0.83)}$ | $75.91_{(\pm0.64)}$ | $\mathbf{79.21}_{(\pm0.69)}$ |
| | Popular | LLaVA1.5 | ✗ | $79.90_{(\pm0.33)}$ | $80.85_{(\pm0.31)}$ | $78.36_{(\pm0.54)}$ | $79.59_{(\pm0.37)}$ |
| | | | ✓ | $\mathbf{82.45}_{(\pm0.30)}$ | $78.26_{(\pm0.38)}$ | $89.91_{(\pm0.31)}$ | $\mathbf{83.68}_{(\pm0.27)}$ |
| | | Qwen-VL | ✗ | $85.56_{(\pm0.35)}$ | $90.44_{(\pm0.56)}$ | $79.53_{(\pm0.84)}$ | $84.63_{(\pm0.42)}$ |
| | | | ✓ | $\mathbf{85.80}_{(\pm0.26)}$ | $85.28_{(\pm0.42)}$ | $86.55_{(\pm0.40)}$ | $\mathbf{85.91}_{(\pm0.25)}$ |
| | | MiniGPT-4 | ✗ | $68.66_{(\pm0.38)}$ | $69.71_{(\pm0.46)}$ | $66.00_{(\pm0.71)}$ | $67.80_{(\pm0.44)}$ |
| | | | ✓ | $\mathbf{75.45}_{(\pm0.63)}$ | $75.14_{(\pm0.72)}$ | $76.09_{(\pm0.70)}$ | $\mathbf{75.61}_{(\pm0.60)}$ |
| | Adversarial | LLaVA1.5 | ✗ | $74.04_{(\pm0.34)}$ | $72.08_{(\pm0.53)}$ | $78.49_{(\pm0.38)}$ | $75.15_{(\pm0.23)}$ |
| | | | ✓ | $\mathbf{75.06}_{(\pm0.18)}$ | $69.24_{(\pm0.27)}$ | $90.20_{(\pm0.53)}$ | $\mathbf{78.34}_{(\pm0.15)}$ |
| | | Qwen-VL | ✗ | $\mathbf{79.57}_{(\pm0.31)}$ | $79.77_{(\pm0.34)}$ | $79.23_{(\pm0.73)}$ | $79.50_{(\pm0.38)}$ |
| | | | ✓ | $78.38_{(\pm0.18)}$ | $74.49_{(\pm0.24)}$ | $86.33_{(\pm0.30)}$ | $\mathbf{79.97}_{(\pm0.15)}$ |
| | | MiniGPT-4 | ✗ | $63.51_{(\pm0.38)}$ | $63.16_{(\pm0.50)}$ | $64.85_{(\pm0.54)}$ | $63.99_{(\pm0.27)}$ |
| | | | ✓ | $\mathbf{70.97}_{(\pm0.24)}$ | $68.80_{(\pm0.19)}$ | $76.72_{(\pm0.55)}$ | $\mathbf{72.55}_{(\pm0.29)}$ |
| GQA | Random | LLaVA1.5 | ✗ | $83.73_{(\pm0.27)}$ | $87.16_{(\pm0.39)}$ | $79.12_{(\pm0.35)}$ | $82.95_{(\pm0.28)}$ |
| | | | ✓ | $\mathbf{86.37}_{(\pm0.07)}$ | $84.86_{(\pm0.24)}$ | $88.58_{(\pm0.41)}$ | $\mathbf{86.68}_{(\pm0.12)}$ |
| | | Qwen-VL | ✗ | $80.97_{(\pm0.32)}$ | $88.07_{(\pm0.34)}$ | $71.64_{(\pm0.57)}$ | $79.01_{(\pm0.40)}$ |
| | | | ✓ | $\mathbf{87.11}_{(\pm0.38)}$ | $89.83_{(\pm0.43)}$ | $83.71_{(\pm0.57)}$ | $\mathbf{86.66}_{(\pm0.41)}$ |
| | | MiniGPT-4 | ✗ | $70.93_{(\pm0.55)}$ | $73.10_{(\pm0.57)}$ | $66.21_{(\pm0.66)}$ | $69.49_{(\pm0.61)}$ |
| | | | ✓ | $\mathbf{80.24}_{(\pm0.19)}$ | $82.96_{(\pm0.35)}$ | $76.12_{(\pm0.90)}$ | $\mathbf{79.39}_{(\pm0.34)}$ |
| | Popular | LLaVA1.5 | ✗ | $78.17_{(\pm0.17)}$ | $77.64_{(\pm0.26)}$ | $79.12_{(\pm0.35)}$ | $78.37_{(\pm0.18)}$ |
| | | | ✓ | $\mathbf{78.91}_{(\pm0.48)}$ | $74.24_{(\pm0.27)}$ | $88.60_{(\pm0.86)}$ | $\mathbf{80.79}_{(\pm0.52)}$ |
| | | Qwen-VL | ✗ | $75.99_{(\pm0.33)}$ | $78.62_{(\pm0.41)}$ | $71.40_{(\pm0.38)}$ | $74.84_{(\pm0.34)}$ |
| | | | ✓ | $\mathbf{81.26}_{(\pm0.38)}$ | $79.82_{(\pm0.38)}$ | $83.68_{(\pm0.39)}$ | $\mathbf{81.70}_{(\pm0.36)}$ |
| | | MiniGPT-4 | ✗ | $65.96_{(\pm0.45)}$ | $65.76_{(\pm0.46)}$ | $66.61_{(\pm1.06)}$ | $66.18_{(\pm0.59)}$ |
| | | | ✓ | $\mathbf{74.40}_{(\pm0.39)}$ | $73.69_{(\pm0.58)}$ | $75.91_{(\pm0.27)}$ | $\mathbf{74.78}_{(\pm0.28)}$ |
| | Adversarial | LLaVA1.5 | ✗ | $\mathbf{75.08}_{(\pm0.33)}$ | $73.19_{(\pm0.49)}$ | $79.16_{(\pm0.35)}$ | $76.06_{(\pm0.24)}$ |
| | | | ✓ | $74.44_{(\pm0.36)}$ | $69.19_{(\pm0.23)}$ | $88.20_{(\pm0.67)}$ | $\mathbf{77.55}_{(\pm0.36)}$ |
| | | Qwen-VL | ✗ | $75.46_{(\pm0.63)}$ | $77.92_{(\pm0.73)}$ | $71.07_{(\pm0.97)}$ | $74.33_{(\pm0.71)}$ |
| | | | ✓ | $\mathbf{79.41}_{(\pm0.41)}$ | $77.04_{(\pm0.61)}$ | $83.81_{(\pm0.73)}$ | $\mathbf{80.28}_{(\pm0.38)}$ |
| | | MiniGPT-4 | ✗ | $62.99_{(\pm0.64)}$ | $62.15_{(\pm0.58)}$ | $66.48_{(\pm0.88)}$ | $64.24_{(\pm0.68)}$ |
| | | | ✓ | $\mathbf{70.60}_{(\pm0.23)}$ | $68.74_{(\pm0.26)}$ | $75.57_{(\pm0.26)}$ | $\mathbf{71.99}_{(\pm0.20)}$ |

Table 11: **Results on POPE.** The best performances for baselines in each setup is highlighted in **bolded**.

| Model | w/ ours | Posters | Celebrity | Scene | Landmark | Artwork | OCR | Perception Total |
|---|---|---|---|---|---|---|---|---|
| LLaVA1.5 | ✗ | $130.14_{\pm3.27}$ | $100.06_{\pm1.52}$ | $144.35_{\pm2.79}$ | $127.70_{\pm2.72}$ | $73.00_{\pm2.70}$ | $99.50_{\pm6.78}$ | $674.74_{\pm8.53}$ |
| | ✓ | $\mathbf{136.12}_{\pm3.60}$ | $\mathbf{116.06}_{\pm4.44}$ | $\mathbf{153.30}_{\pm2.72}$ | $\mathbf{140.05}_{\pm2.92}$ | $\mathbf{101.25}_{\pm3.29}$ | $\mathbf{101.00}_{\pm8.15}$ | $\mathbf{747.78}_{\pm11.16}$ |
| Qwen-VL | ✗ | $148.19_{\pm3.85}$ | $117.79_{\pm2.95}$ | $158.75_{\pm1.68}$ | $147.42_{\pm3.67}$ | $115.50_{\pm3.38}$ | $86.25_{\pm11.79}$ | $773.90_{\pm10.06}$ |
| | ✓ | $\mathbf{165.48}_{\pm0.70}$ | $\mathbf{126.62}_{\pm0.50}$ | $\mathbf{164.00}_{\pm2.48}$ | $\mathbf{153.63}_{\pm2.01}$ | $\mathbf{129.50}_{\pm2.72}$ | $\mathbf{87.50}_{\pm14.14}$ | $\mathbf{826.72}_{\pm9.44}$ |

Table 12: Results on all MME perception-related tasks. The best performance of each setting is **bolded**.

| Model | w/ ours | Common Sense Reasoning | Numerical Calculation | Text Translation | Code Reasoning | Recognition Total |
|---|---|---|---|---|---|---|
| LLaVA1.5 | ✗ | $52.86_{\pm6.28}$ | $50.00_{\pm8.51}$ | $17.50_{\pm12.35}$ | $44.00_{\pm11.02}$ | $164.36_{\pm20.16}$ |
| | ✓ | $\mathbf{97.71}_{\pm7.32}$ | $\mathbf{57.50}_{\pm11.07}$ | $\mathbf{74.00}_{\pm9.30}$ | $\mathbf{68.00}_{\pm10.30}$ | $\mathbf{297.21}_{\pm9.61}$ |
| Qwen-VL | ✗ | $122.74_{\pm4.92}$ | $49.58_{\pm10.94}$ | $121.25_{\pm7.47}$ | $73.75_{\pm13.90}$ | $367.32_{\pm23.43}$ |
| | ✓ | $\mathbf{126.79}_{\pm7.57}$ | $\mathbf{58.75}_{\pm14.56}$ | $\mathbf{139.17}_{\pm16.62}$ | $\mathbf{76.67}_{\pm14.48}$ | $\mathbf{401.37}_{\pm30.64}$ |

Table 13: Results on all MME cognition-related tasks. The best performance of each setting is **bolded**.

and OCRVQA [48], as the training data. Similarly, Qwen-VL-Chat [2] is trained using datasets that

include VQAv2, GQA, and OCRVQA. For MiniGPT-4 [78], we adopt the same datasets used for LLaVAv1.5-7b. The corresponding hyperparameters are summarized in Table 10.

### D.2 Additional Experiments on POPE

We present a comprehensive performance evaluation of our VLR-distillation method applied to POPE, across two additional datasets: A-OKVQA and GQA. As shown in Table 11, our approach outperforms the baselines across nearly all configurations, particularly when compared to the MiniGPT-4 baseline, with an average improvement of $7.3\%$ in accuracy and $8.4\%$ in F1 score. Furthermore, we observe significant improvements for LLaVAv1.5 on A-OKVQA and for Qwen-VL on GQA, with average gains on accuracy of $2.6\%$ and $5.12\%$, respectively.

### D.3 Additional Experiments on MME Remaining Subset

We evaluate the performance of our proposed method on the MME remaining set, with results presented in Tables 12 and 13. Table 12 shows the performance of the perception-related tasks, while Table 13 focuses on the cognition-related tasks. Our method consistently outperforms the three baseline approaches across both perception and cognition tasks. Notably, it exhibits significant improvements in cognitive performance, which we attribute to its effective handling of SCHall—potentially a key factor influencing cognition-related tasks in the MME dataset.

## E Visualizations

We provide visualizations of our SCBench benchmark in Figure 9 and a demonstration of the effectiveness of our VLR-distillation in Figure 10.

Specifically, we present representative samples for each category in our SCBench benchmark, as shown in Figure 9. It can be observed that the images and questions in our benchmark are not particularly challenging for current powerful LVLMs. However, these models still struggle to answer the questions. Besides, Figure 10 illustrates the effectiveness of our method on each category in our SCBench, with each background color corresponding to a distinct category in the benchmark.

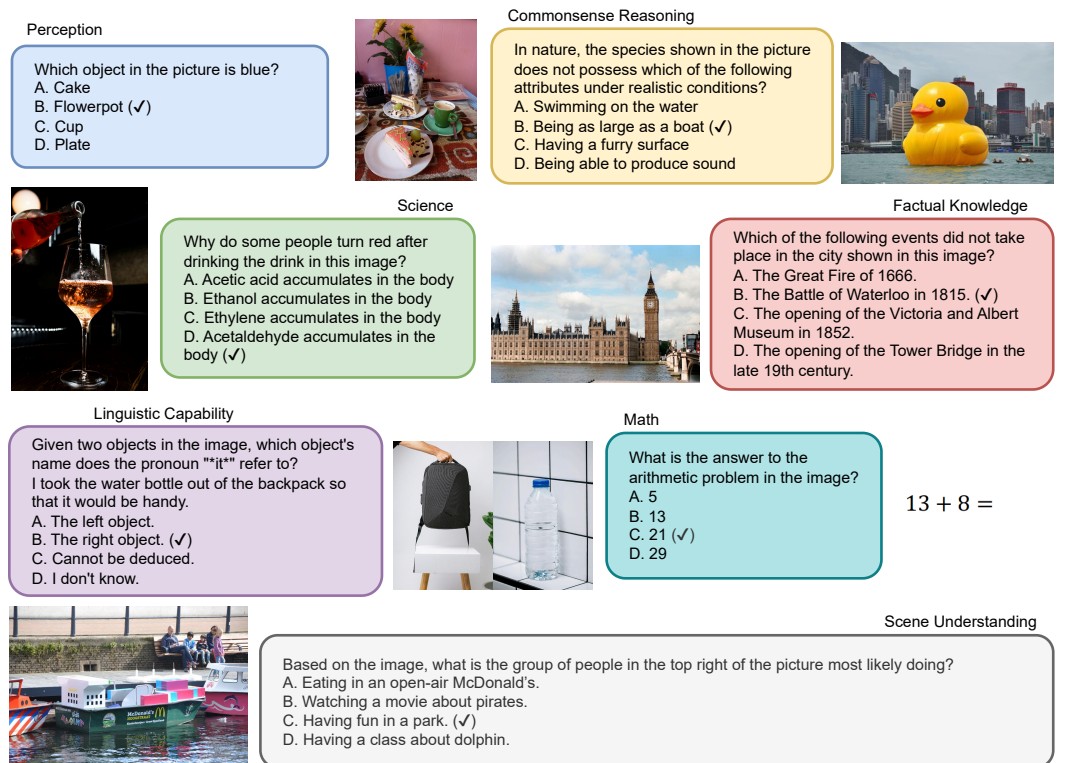

Figure 9: Visualizations of questions in SCBench Benchmark. Our benchmark considers both visual- and textual-centric tasks which are likely to induce SCHall. The ground-truth answer for each question is indicated with a ✓.

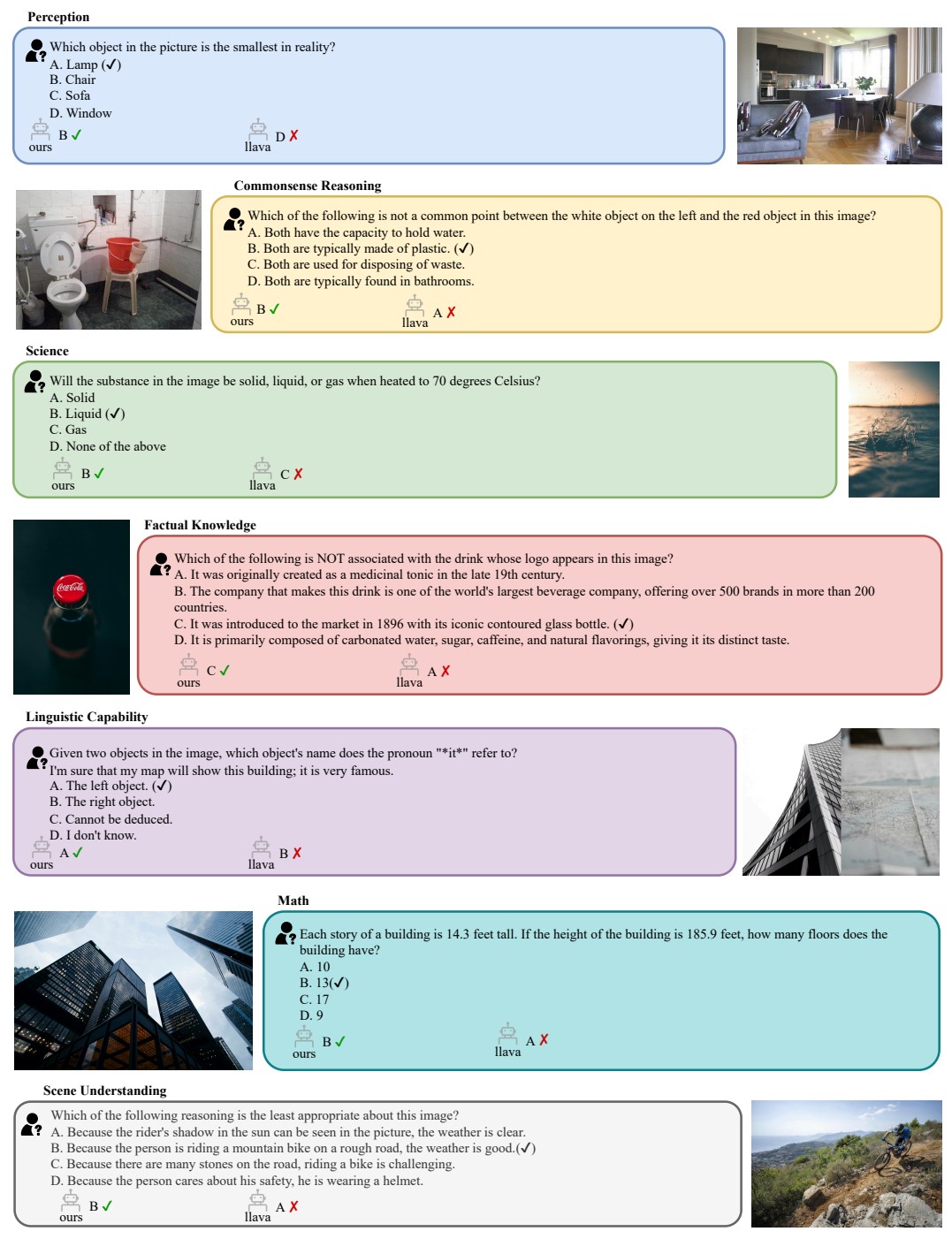

Figure 10: Visualizations of the effectiveness of our VLR-distillation method in SCBench Benchmark.

