# OpenReview forum: "Discovering Compositional Hallucinations in LVLMs"
_NeurIPS.cc/2025/Conference — NeurIPS 2025 poster_

### Official Review · Reviewer_riVB · 2025-06-19

**Clarity:** 2
**Significance:** 2
**Originality:** 3
**Rating:** 3
**Confidence:** 4

**Summary:**

This paper introduces a hallucination benchmark, Simple Compositional Hallucination (SCHall), constructed from the compositional structures of text- and vision-centric questions. Through a comprehensive analysis of the challenges posed by compositional hallucinations, the authors further propose a method called VLR-Distillation, which distills knowledge from a large language model (LLM) augmented with additional image captions. Experimental results on both hallucination benchmarks and general visual question answering (VQA) benchmarks demonstrate the effectiveness of the proposed approach.

**Questions:**

N/A

**Ethical Concerns:**

["NO or VERY MINOR ethics concerns only"]

**Final Justification:**

I have carefully read the authors’ rebuttal. However, I still do not clearly understand the key differences between the proposed benchmark and existing commonsense-based VQA datasets, as both require models to extract information from the input image and question. Moreover, the distinction between hallucination and traditional VQA errors remains unclear, making it difficult to understand why this should be considered a hallucination benchmark. So I will keep my score.

**Limitations:**

yes

**Quality:**

2

**Strengths And Weaknesses:**

Strengths
1. The paper presents a thoughtful analysis of the challenges and underlying causes of compositional hallucination in multimodal reasoning tasks.
2. The proposed method is thoroughly evaluated on both hallucination-specific and general VQA benchmarks, demonstrating its effectiveness and versatility.

Weakness
1. The distinction between the proposed SCHall benchmark and existing common-sense knowledge-based VQA benchmarks (e.g., OK-VQA, AOK-VQA) remains unclear. Based on Fig. 1 and the examples provided in the Appendix, the proposed benchmark appears largely similar to previous ones. As such, the use of the term hallucination may come across as more of a rebranding than a substantive contribution.
2. The scale of the SCHall benchmark is relatively limited, raising concerns about its representativeness and generalizability.
3. The paper’s writing and logical flow require improvement to enhance readability and clearly convey the motivation and contributions.

---

> ### Author Rebuttal · Authors · 2025-07-31
>
> We thank the reviewer for the feedbacks and questions. Our response is as follows:
>
> > **W1:** Distinction between the proposed benchmark and existing VQA benchmarks
> >
>
> We would like to clarify that our proposed VLComp benchmark differs fundamentally from existing knowledge-based VQA benchmarks. As stated in Lines 56–59 in the main paper, while challenges related to **textual knowledge** can arise in these VQA benchmarks, VLComp focuses on questions with **simpler textual sub-questions**, with its primary challenge being ***compositional hallucination*.**
> To further emphasize this distinction, we present the following comparison table as follows:
>
> |Benchmark|QuestionType|Visual|||Lingual|Easy Visual|Easy Lingual|Compositional|
> | ------------- | ------------- | ------ | --------- | -------- | ------- | --- | --- | --- |
> |||Object|Attribute|Relation|||||
> |OK-VQA|VQA|✔|✔|✔|✔|✘|✘|✔|
> |A-OKVQA|VQA,MCQ|✔|✔|✔|✔|✘|✘|✔|
> |POPE|Y/N|✔|✘|✘|✘|✘|✘|✘|
> |MME|Y/N|✔|✔|✘|✘|✘|✘|✘|
> |CHAIR|Open|✔|✘|✘|✘|✘|✘|✘|
> |AMBER|Open,Y/N|✔|✔|✔|✘|✘|✘|✘|
> |MMHal-Bench|VQA|✔|✘|✘|✘|✘|✘|✘|
> |HaluEval|QA,Short|✘|✘|✘|✔|✘|✘|✘|
> |DiaHalu|Short|✘|✘|✘|✔|✘|✘|✘|
> |VLHall(Ours)|MCQ|✔|✔|✔|✔|✔|✔|✔|
>
>
> > **W2:** Concerns about VLComp scale.
> >
>
> The representativeness and generalizability of VLComp have been validated from the following aspects:
>
> - VLComp is specifically designed to diagnose and analyze SCHall. Inspired by existing hallucination evaluation benchmarks for LVLMs—such as LLaVA-Bench-in-the-Wild, MMHal-Bench, and MM-Vet, which comprise approximately 60, 96, and 128 samples, respectively—VLComp significantly expands both scale (323 samples with 951 compositional and decomposed questions in total) and scope. Importantly, it ensures strong diversity across key categories, including perception, science, commonsense reasoning, factual knowledge, linguistic capability, scene understanding, and mathematics.
> - While VLComp is relatively small than more general-purpose LVLM benchmarks, our experiments show that it effectively identifies and supports exploration of this novel form of hallucination. **More importantly, SCHall also emerge in general-purpose benchmarks (Figure2)**, suggesting that our benchmark captures a representative and generalizable issue.
> - In addition, the semi-automated process for constructing VLComp involves minimal manual annotation, primarily limited to reviewing and refining the questions. This design allows for efficient and scalable expansion. In future work, we plan to extend VLComp with broader coverage and increased human involvement to further enhance its reliability and generality.
>
> > **W3:** Concerns on writing and logical flow.
> >
>
> We would like to clarify that our paper is structured to follow a logical progression centered around a concrete motivation: the discovery of a previously underexplored hallucination type in LVLMs, which we term SCHall (Simple Compositional Hallucination). The paper is organized as follows:
>
> 1. We define SCHall and motivate its importance based on empirical observations (Introduction & Section 3.1).
> 2. We construct VLComp, a targeted benchmark to systematically study this phenomenon (Section 2).
> 3. We analyze its underlying causes, identifying two contributing factors: (Section 3.2 & 3.3)
> (i) difficulty with visual abstraction, and
> (ii) interference of visual input with language reasoning.
> 4. We then propose VLR-distillation, a mitigation method informed by our analysis (Section 4).
>
> In the final version, we will improve the transitions between sections to better highlight the logical flow and the relationship between sections.

---

> ### Author Response · Authors · 2025-08-04
>
> Thank you for your response and for engaging in the discussion! We highlight the fundamental differences between our compositional hallucination benchmark and OK-VQA, as well as the differing treatments of hallucination and error in this study, as follows:
>
> ---
>
> **Our compositional hallucination benchmark vs. OK-VQA**
>
> We would like to clarify that our work is centered on the proposal and investigation of ***Simple Compositional Hallucination (SCHall)***, and the benchmark we introduce is purposefully designed to support this goal. The key distinction between our benchmark and existing ones is that ours is **uniquely capable of revealing compositional hallucinations in a significant and systematic way**. For instance, the OK-VQA benchmark is not suitable for analyzing SCHall, for the following reasons:
>
> 1. **Lack of Compositionality in OK-VQA:** OK-VQA is not specifically designed to evaluate compositionality but rather focuses on knowledge-based question answering. **Neither the data construction process nor the evaluation in OK-VQA takes compositionality into account.** As a result, a significant portion of its questions lack compositional structure. For instance, questions like "What toy is this?" are included in OK-VQA because their answers (e.g., "stuffed animal", "teddy bear") require world knowledge. However, the type of compositionality we focus on generally involves combining and reasoning over recognized visual elements in a structured manner. Moreover, in OK-VQA, some questions such as "What do they call running around the bases on a single hit?" can be answered purely based on knowledge without any visual input. Due to time constraints, we analyze the first 100 questions in the OK-VQA’s validation set and find that **59% are non-compositional in OK-VQA**: 49% are direct knowledge-based recognition questions, and 10% could be answered without visual information at all.
> 2. **The Performance Bottleneck in OK-VQA Is Not Due to Compositionality:** Unlike ours, the performance bottleneck in OK-VQA does not stem from compositionality, but rather from the model’s limitations in external knowledge. To illustrate this, we decompose the all remaining compositional problems in the OK-VQA subset into visual and textual subproblems, following the same procedure as in our benchmark. The results reveal accuracies of 93.6% for visual subproblems, 73.2% for textual subproblems, and 70.7% for the original task—similar to the textual subproblems. **This indicates that the bottleneck lies in the textual subproblems, which, for OK-VQA, corresponds to limitations in external knowledge.** This phenomenon differs from what we observe in our SCHall benchmark **(Figure 1c)** and general-purpose LVLM benchmarks **(Figure 2)**. One possible reason is that the OK-VQA has been extensively used to train LVLMs (e.g., LLaVA 1.5), and thus it may not effectively reflect SCHall during evaluation.
>
> **Error vs. Hallucination**
>
> **Errors** broadly refer to any incorrect model outputs, which can result from missing knowledge, reasoning failures, instruction-following misalignment, hallucinations, or other limitations. In this study, we adopt a more precise definition to differentiate hallucinations from general errors. Specifically, **errors** are defined as failures caused by insufficient capabilities (i.e., missing knowledge or recognition errors). In contrast, **hallucinations** occur **when the model possesses the necessary knowledge and recognition capabilities to answer the question correctly, yet still produces a response that is inconsistent with the input or factual reality.** Accordingly, **SCHall** is considered a hallucination rather than a general error: while the model successfully solves the subproblems it is capable of answering—demonstrating both the required knowledge and recognition capabilities—it hallucinates when addressing compositional questions derived from them. Whereas benchmarks like **OK-VQA** primarily capture errors arising from knowledge gaps, our benchmark is specifically designed to target **SCHall**.
>
> Thanks for your feedback again. We welcome further discussion should you have any questions or require additional clarification.

---

> > ### Comment · Reviewer_riVB · 2025-08-05
> >
> > Thank you for your responses. I agree that prior VQA datasets were not specifically designed to target compositional hallucinations. However, as the authors acknowledge, 41% of existing data in OK-VQA is compositional, indicating that compositional VQA is not an entirely new problem.
> > While I recognize the value of a dedicated compositional evaluation, the relatively limited scale of the dataset and its partial overlap with existing benchmarks constrain the overall contribution of this work. Therefore, I have decided to raise my score to 3. I recommend that the authors consider incorporating compositional samples from existing datasets to further expand and enhance their benchmark.

---

> ### Author Response · Authors · 2025-08-05
>
> We sincerely thank you for raising the score and for the timely discussion! As you have agreed, prior VQA datasets were not tailored for compositional hallucinations. In addition, we would like to gently point out that datasets such as OK-VQA **are not only not specifically designed with this focus in mind, but also are less well-suited for evaluating compositionality.** This is mainly because the accuracy on original OK-VQA questions closely matches that on their text-centric sub-questions, indicating that **OK-VQA’s main challenge lies in external knowledge within the textual sub-questions rather than compositionality.** Consequently, despite 41% of questions being decomposable, OK-VQA is not well-suited for studying compositionality.
>
> As our benchmark, covering diverse fields, identifies compositional hallucination issues as significant (Figure 1c), and given that these issues are also prevalent in general benchmarks (Figure 2), we respectfully believe that the current benchmark **is sufficient for investigating and evaluating compositional hallucinations**. That said, we agree that scaling up—whether through our semi-automatic method or by first diagnosing existing benchmarks and extracting subsets containing compositional hallucinations—would further strengthen our claims. Nonetheless, **we view this as an enhancement and expansion, rather than an essential prerequisite.** Specifically, since the benchmark represent **only part of our contribution**, the main contribution of our paper also lies in **being the first to identify, define, and study compositional hallucinations, as well as proposing effective novel solutions.**
>
> Once again, we sincerely thank you for raising the score, providing the timely discussion, and offering the valuable perspective on potential scaling!

---

### Official Review · Reviewer_4hUj · 2025-07-02

**Clarity:** 3
**Significance:** 3
**Originality:** 3
**Rating:** 4
**Confidence:** 4

**Summary:**

The paper introduces Simple Compositional Hallucination (SCHall) — a novel and subtle failure mode in large vision-language models (LVLMs). SCHall arises not from visual or textual input alone, but from their composition, where individually “easy” visual and textual questions—answered correctly in isolation—lead to hallucinated answers when combined.

To systematically study this issue, the authors build a benchmark called VLComp, composed of question triplets: simple visual, simple textual, and their compositional form. Surprisingly, strong LVLMs like GPT-4o, InternVL, and LLaVA exhibit up to a 20% drop in accuracy when moving from isolated to compositional questions.

The authors identify two contributing factors: (1) failure in abstracting and aligning relevant visual content during composition, and (2) degradation in language processing performance when visual inputs are present. To address this, they propose VLR-distillation, a novel training strategy that incorporates Vision Language Registers (VLRs) and a distillation branch to enhance visual grounding and preserve linguistic reasoning.

Extensive experiments show their method significantly reduces SCHall and improves performance across multiple hallucination and VQA benchmarks, demonstrating the generalizability of their findings.

**Questions:**

1. What types of compositional hallucinations are most common, and how do different models fail in different ways?
2. You attribute SCHall to failures in visual abstraction and degradation of language processing. However, how do you distinguish SCHall from general shortcut learning or heuristic answering strategies that might also cause similar issues?
3. My other questions are mentioned in the weakness. I am willing to raise my score if the authors can answer my questions.

**Ethical Concerns:**

["NO or VERY MINOR ethics concerns only"]

**Final Justification:**

My concerns are addressed so I will maintain my positive score.

**Limitations:**

yes

**Quality:**

3

**Strengths And Weaknesses:**

**Strength**

1. The paper identifies and formalizes a new type of hallucination in LVLMs—Simple Compositional Hallucination (SCHall)—which occurs even when both visual and textual components are individually handled correctly. This is a subtle, underexplored, and practically relevant failure mode that has not been systematically studied before.

2. The authors introduce VLComp, a carefully curated benchmark that tests LVLMs' robustness to compositional queries. The benchmark is well-constructed using a mix of automatic generation and human filtering, with a diverse set of visual and textual question types.

3. The paper presents compelling evidence that stronger models (e.g., GPT-4o, Qwen-VL) exhibit more pronounced SCHall, implying that current progress on visual recognition alone is insufficient.

4. The proposed Vision Language Registers (VLRs) and distillation learning strategy are elegant, practically implementable solutions.

**Weakness**

1. While VLComp is well-motivated, it is relatively small and semi-automatically constructed, which may raise questions about generalizability and annotation bias.

2. While VLR-distillation performs well, the paper provides limited comparative insight into why existing mitigation strategies (like REVERIE or PAI) fail in this specific scenario. A qualitative failure case analysis or sensitivity study could enhance interpretability.

3. Can the author evaluate the method on more Hallucination evaluation benchmark like HallusionBench? What about other VQA benchmarks like MMMU, MathVista?

---

> ### Author Rebuttal · Authors · 2025-07-31
>
> We thank the reviewer for the insightful feedbacks and questions. Our response is as follows:
>
> > **W1:** Concerns about generalizability and annotation bias.
> >
>
> Thank you for your feedback. The generalizability and quality of VLComp can be validated from the following aspects:
>
> - VLComp is specifically designed to diagnose and analyze SCHall. Inspired by existing hallucination evaluation benchmarks for LVLMs—such as LLaVA-Bench-in-the-Wild, MMHal-Bench, and MM-Vet, which comprise approximately 60, 96, and 128 samples, respectively—VLComp significantly expands both scale (323 samples with 951 compositional and decomposed questions in total) and scope. Importantly, it ensures strong diversity across key categories, including perception, science, commonsense reasoning, factual knowledge, linguistic capability, scene understanding, and mathematics.
> - While VLComp is relatively small than more general-purpose LVLM benchmarks, our experiments show that it effectively identifies and supports exploration of this novel form of hallucination. **More importantly, SCHall also emerge in general-purpose benchmarks** (Figure2), suggesting that our benchmark captures a representative and generalizable issue.
> - In addition, the semi-automated process for constructing VLComp involves minimal manual annotation, primarily limited to reviewing and refining the questions. This design allows for efficient and scalable expansion. In future work, we plan to extend VLComp with broader coverage and increased human involvement to further enhance its reliability and generality.
>
> > **W2:** Interpretation on failure of existing hallucination mitigation strategies on SCHall.
> >
>
> Thank you for the insightful comment. The ineffectiveness of existing mitigation strategies in addressing SCHall can be attributed to two primary factors:
>
> 1. While methods like PAI are effective at mitigating object hallucination and improving recognition accuracy, they are less suited to the SCHall setting, which primarily involves compositional reasoning rather than recognition failures. For instance, in Figure 5 (Math) of the Appendix, the image displays the equation “13 + 8,” which the baseline model answers correctly. However, PAI misleads the model to predict “53”—a partially visually plausible (due to the “3”) but semantically incorrect answer. This highlights that the difficulty in SCHall stems more from compositional challenge than from visual recognition errors.
> 2. Meanwhile, REVERIE attributes hallucinations to a lack of fine-grained supervision and improves reasoning via Reflective Instruction Tuning. However, it does not explicitly teach selective integration of visual and textual cues. For instance, as illustrated in Figure 3(a) in Appendix, where a person is clearly chopping wood at sunset, the baseline model incorrectly predict "thermal energy" based on the visual salience of sunlight. REVERIE fails to correct this, likely because both the ground-truth and hallucinated answers are visually plausible.
>
> We appreciate the suggestion and will include additional qualitative failure cases and comparative analyses for existing methods in the final version to enhance interpretability.
>
> > **W3:** More evaluation results on hallucination and other VQA benchmarks.
> >
>
> Thank you for your suggestion! We evaluate our method with LLaVA-1.5-7B on additional hallucination benchmarks, including HallusionBench, as well as general-purpose VQA datasets, including MMBench, ScienceQA, and MM-Vet. Results are shown below.
>
> |  | MMBench | ScienceQA | MM-Vet | Hallusion Bench |
> | --- | --- | --- | --- | --- |
> | LLaVA-v1.5-7b | 64.3 | 66.8 | 31.1 | 47.6 |
> | + ours | 65.4 | 67.8 | 33.3 | 49.5 |
>
> Our method achieves consistent improvements across multiple benchmarks, demonstrating both the generality of SCHall and the effectiveness of our approach.
>
> > **Q1:** Types of compositional hallucinations and how do models fail in them?
> >
>
> Thank you for your valuable suggestion! We identify several common types of **compositional hallucinations** observed in our VLComp:
>
> 1. **Irrelevant Visual Feature Attribution.** When an image contains rich or distracting content, the model may attend to semantically irrelevant visual regions, leading to incorrect reasoning. For instance, in Figure 3a in Appendix, the image shows a person chopping wood at sunset, while the question concerns the type of energy consumed during the activity. Earlier models (e.g., LLaVA) are often misled by salient but contextually unrelated elements (e.g., the sunset), resulting in hallucinated answers.
> 2. **Superficial Visual-Linguistic Mapping.** When there is superficial alignment between visual content and candidate answers, models may rely on pattern matching instead of grounded reasoning.
> In Figure 3b, a dog is resting on a chair. The question asks about *typical behaviors* of the animal shown. Most open-source models mistakenly select “being very lazy” based on the dog’s current state, rather than reasoning about its typical behavior in general.
> 3. **Cross-modal Misalignment.** In some cases, replacing the image with its caption leads to correct answers, indicating poor alignment between visual and textual modalities. In Figure 4 in appendix, the image contains a clearly visible ruler, and the question asks what it can be used to measure. While caption-based inputs (e.g., “A ruler is shown”) yield correct responses, the same models fail with the full visual input. Such failures on simple and unambiguous images are typically observed only in earlier models (e.g., LLaVA).
>
> Nevertheless, the interplay of these challenges often leads to failure cases that challenge even close-source models. There remain many types of compositional hallucinations that warrant deeper investigation. We will incorporate the above discussion and related analyses in the final version.
>
> > **Q2:** Discussion with general shortcut learning or heuristic answering strategies.
> >
>
> We thank the reviewer for the thoughtful question. We agree that shortcut learning and heuristic answering are important considerations and likely contribute to the observed SCHall behaviors, particularly those related to *Superficial Visual-Linguistic Mapping* (see response to Q1).
>
> However, we argue that SCHall extends beyond these mechanisms. In addition to superficial patterns, SCHall captures deeper model limitations such as *Irrelevant Visual Feature Attribution* and *Cross-modal Misalignment.* These cannot be fully accounted for by shortcut learning alone. We view our analysis as an initial step, and we welcome future work that further disentangles these contributing factors.

---

> > ### Comment · Reviewer_4hUj · 2025-08-04
> >
> > Thank you for the detailed response. Most of my concerns have been addressed, and I therefore maintain my score. I have one more question. On MMBench, ScienceQA, MM-Vet and HallusionBench benchmarks, can you also the show the results of 13B models or other models instead of LLaVA-v1.5-7b?

---

> > > ### Author Response · Authors · 2025-08-05
> > >
> > > Thank you for your response! We are glad that our rebuttal has addressed most of your concerns. Following your suggestion, we have evaluated our method using Qwen-VL and MiniGPT-4 (i.e., the same model referenced in Table 2, 3 and 4) on the MMBench, ScienceQA, MM-Vet, and HallusionBench benchmarks. The results, presented below, demonstrate consistent improvements.
> > >
> > > |  | MMBench | ScienceQA | MM-Vet | Hallusion Bench |
> > > | --- | --- | --- | --- | --- |
> > > | Qwen-VL-Chat | 60.6 | 68.2 | 40.8 | 57.0 |
> > > | + ours | 64.6 | 69.1 | 41.4 | 57.8 |
> > > | MiniGPT-4 | 35.7 | 49.9 | 21.6 | 44.4 |
> > > | + ours | 36.6 | 52.8 | 23.1 | 50.1 |
> > >
> > > Please do not hesitate to reach out if you have any further questions or require additional clarification. Thank you once again!

---

### Official Review · Reviewer_cKHX · 2025-07-02

**Clarity:** 3
**Significance:** 3
**Originality:** 3
**Rating:** 4
**Confidence:** 4

**Summary:**

This paper investigates a previously underexplored type of hallucination in large vision-language models (LVLMs), termed Simple Compositional Hallucination (SCHall). This form of compositional questioning consistently triggers hallucinations, despite the models achieving high accuracy on the decomposed sub-questions when handled independently.

To systematically study this phenomenon, the authors construct a new benchmark called  VLComp, which comprises triplets of questions: vision-only, text-only, and their compositional combination. They demonstrate that SCHall is a widespread issue and becomes more pronounced in models with stronger visual recognition capabilities.

To mitigate SCHall, the authors propose VLR-Distillation, a novel framework that introduces Vision Language Registers (VLRs) along with a textual distillation training objective, aiming to preserve language modeling capacity under visual conditions. Experimental results show consistent improvements over several strong baselines on the proposed VLComp benchmark, as well as on other hallucination or general-purpose VQA datasets.

**Questions:**

1. Which model is applied for captioner?
2. In the ablation study, how many VLRs were used in the with/without DL (distillation learning) settings? Could increasing the number of VLRs (e.g., setting it to 8) achieve similar performance to using distillation learning?
3. In the ablation study, why does increasing the number of VLRs lead to degraded performance?
4. What are the key differences between your VLRs and LLaVA-Mini[1]?


[1] Zhang, Shaolei, et al. LLaVA-Mini: Efficient Image and Video Large Multimodal Models with One Vision Token. arXiv:2501.03895, arXiv, 2 Mar. 2025. arXiv.org, https://doi.org/10.48550/arXiv.2501.03895.

**Ethical Concerns:**

["NO or VERY MINOR ethics concerns only"]

**Final Justification:**

The paper seems novel but still lack of theoretical analysis. I will keep the rating.

**Limitations:**

yes

**Quality:**

4

**Strengths And Weaknesses:**

Strengths
1. The paper is clearly written and well-structured. The motivation is well-explained with illustrative examples (e.g., Fig. 1), and the experimental setup is carefully detailed.
2. While the paper provides solid empirical evidence, it lacks a deeper theoretical analysis or formalization of why compositionality causes hallucination at the model architecture or optimization level. The explanations are plausible but remain largely observational.
3. The paper addresses a previously unrecognized and important phenomenon—Simple Compositional Hallucination (SCHall)—that affects the reliability of modern vision-language models. Given the increasing deployment of LVLMs in real-world scenarios, understanding and mitigating subtle failure modes like SCHall is of practical and theoretical significance.
4. The identification and formalization of SCHall is novel. The idea that hallucinations can emerge only when combining independently “easy” visual and textual components is counterintuitive and has not been highlighted in prior work. The proposed VLComp benchmark also offers a novel way to evaluate this specific type of hallucination, distinct from existing hallucination datasets.

Weaknesses
1. There are some writing mistakes, such as a formula error in the causal attention mask in Section 4.1.
2. While the paper provides solid empirical evidence, it lacks a deeper theoretical analysis or formalization of why compositionality causes hallucination at the model architecture or optimization level. The explanations are plausible but remain largely observational.

---

> ### Author Rebuttal · Authors · 2025-07-31
>
> We thank the reviewer for the insightful feedbacks and questions. Our response is as follows:
>
> > **W1:** Writing mistakes in Section 4.1
> >
>
> Thank you for pointing out the formatting issues. We will ensure that the correct condition $s \leq r$ is clearly stated and carefully revise the formatting in the final version.
>
> > **W2:** Lacks of deeper theoretical analysis
> >
>
> Indeed, our analysis of SCHall is primarily empirical. We perform error analysis on both VLComp and general benchmarks to reveal its widespread presence, and use experiments and visualizations for preliminary attribution. The competitive performance of our method on VLComp further substantiates our analysis.
>
> We acknowledge the need for deeper theoretical analysis of why compositionality leads to hallucination at the model architecture and optimization level. Our hypothesis is that the training paradim, which supervises only the final textual output without intermediate guidance, leads the model to rely on spurious correlations and dataset biases instead of grounded compositional reasoning, thereby increasing SCHall risk. In response to this suggestion, we plan to conduct further analyses, including intermediate representation probing with logit lens and intervention validation via attention transfer.
>
> > **Q1-3:** Experimental Setting Details
> >
>
> Thank you for your question. During training, in cases where ground-truth captions are not available, we employ the baseline model itself as the caption generator. In the ablation study (left table of Table 5), we used 4 VLRs for both the with and without distillation learning (DL) settings. We further conduct additional experiments by increasing the number of VLRs without applying distillation learning (DL). The results are shown below:
>
> | #VLRs | Random Acc | Random F1 | Popular Acc | Popular F1 | Adversarial Acc | Adversarial F1 |
> | --- | --- | --- | --- | --- | --- | --- |
> | 4 | 86.5 | 85.9 | 84.1 | 83.7 | 80.2 | 80.4 |
> | 8 | 86.7 | 86.4 | 83.5 | 83.5 | 79.3 | 80.1 |
> | 16 | 85.0 | 84.9 | 82.0 | 82.4 | 78.4 | 79.7 |
>
> These results indicate that simply increasing the number of VLRs does not match the performance achieved when combining VLRs and DL. Notably, performance even degrades when using 16 VLRs. This may be related to the nature of the POPE benchmark, where 4 tokens are sufficient to capture the necessary visual information. Adding more VLRs may introduce redundancy and noise, which could lead to hallucinations—this also addresses the concern raised in Q3.
>
> > **Q4:** Key differences with LLaVA-Mini
> >
>
> Thank you for your question. LLaVA-Mini employs **explicitly designed fusion and compression modules** to generate compressed and **pre-fusion tokens**. In contrast, our approach introduces VLRs, a novel token type that operates entirely **within the LVLM,** enabling it to autonomously learn to extract question-relevant image information while also engaging in textual understanding. Specifcally, the key differences are as follows:
>
> - Our VLRs are independent tokens that bridge vision and language, whereas LLaVA-Mini uses pre-fusion tokens to replace text tokens.
> - The multimodal interaction in our method is conditioned on the LVLM’s rich internal knowledge, while LLaVA-Mini performs a simpler fusion of modalities. By leveraging this internal knowledge, VLRs achieve more flexible and semantically meaningful multimodal representations.

---

### Official Review · Reviewer_r87F · 2025-07-07

**Clarity:** 3
**Significance:** 3
**Originality:** 3
**Rating:** 6
**Confidence:** 5

**Summary:**

This work identifies and analyzes a fundamental hallucination phenomenon in multimodal large models, termed SCHall. To address this, this work introduce the VLComp benchmark for evaluating SCHall and conduct a comprehensive assessment of existing models and mitigation techniques, revealing their significant limitations. Based on the findings, this work propose VLR-distillation, which achieves substantial improvements across VLComp, multiple hallucination benchmarks, and general VQA tasks.

**Questions:**

1. What is the relationship between the evaluation angle proposed in the article and the multimodal alignment problem? It is natural for the model to have better accuracy on a single mode, so are there other angles and views on the combined modal challenge proposed in the article compared to multimodal alignment? From Figure 1, the effect of simple-text is lower than that of simple-visual. Is there a corresponding explanation?

2. From another perspective, does the solution proposed in the article make the multimodal alignment of mllms perform better? I still want to ask the author to introduce the motivations and effects of the three stages in the method part.

**Ethical Concerns:**

["NO or VERY MINOR ethics concerns only"]

**Final Justification:**

- The author has detailed additional statistical analysis experiments for the benchmark.
- The author also supplemented the analysis experiments for the currently mainstream closed-source MLM framework, and obtained conclusions consistent with those in the article.
- The updated methodology includes the latest Qwen2.5-VL model and detailed evaluation experiments.
- The article addresses most of the questions and writing issues.

The article provides a novel perspective on constructing an MLM benchmark, providing the community with new conclusions and findings.

**Limitations:**

- Benchmark:
1. The appendix or the main text lacks a more detailed description of the statistical information of the dataset, as well as the evaluation process and indicators
2. The evaluation on the current mainstream open source and closed source models or the evaluation by human experts to verify the rationality of the benchmark

- Method:
1. The motivation description of the three stages proposed
2. Experiments on more types and sizes of open source models to verify the universality of the method

If the author can solve most of the above problems, I am willing to reconsider the contribution of the article

**Quality:**

3

**Strengths And Weaknesses:**

Strengths：
- The evaluation angle of MLLMs proposed in the article is novel
-  This work proposes an effective solution and baseline based on the problem
- The description of motivation and conclusion is very clear, and the writing is very logical

Weakness:
- The statistical information and construction process of the benchmark need to be introduced in more detail (with the help of charts, etc.)
- There is a lack of evaluation on the current mainstream mllms (including open source and closed source models) to verify the rationality of the motivation and the quality of the benchmark
- In the method part, the mllms used is a bit outdated

---

> ### Author Rebuttal · Authors · 2025-07-31
>
> Thank you for the valuable feedbacks and suggestions. Our response is as follows:
>
> > **W1 & Limitations 1.1:** Statistical information and construction process of the benchmark.
> >
>
> We appreciate your suggestion to provide more detailed information about the statistical properties and construction process of our benchmark.
>
> 1. **Statistical Information**:
>
>     We provided detailed statistical information in the supplementary materials (Appendix C.1 and C.2), including data sources, data volume (951 total questions with 323 compositional and 628 decomposed), and category distribution (18% Perception, 15.2% Science, 12.1% Commonsense Reasoning, 9.3% Factual Knowledge, 9.3% Language Capability, 18.6% Scene Understanding and 17.6% Math). We will include more details (e.g. sub-problems distributions) and make sure to reference these clearly in the main text for better accessibility.
>
> 2. **Benchmark Construction Process**:
>
>     We agree that a clearer explanation of the benchmark construction process would improve the clarity of the paper. Our benchmark construction process includes two stages: Atomic Questions Generation and Simple-Composed Question Generation.
>
>     **Atomic Question Generation**
>
>     - **Image Collection**: Images are collected from datasets (MMBench, MME, SEEDBench) and online sources.
>     - **Caption Generation**: LVLMs (LLaVA, Qwen-VL, MiniGPT-4, InternVL) generate captions for each image.
>     - **Visual-centric Question Formulation:** GPT-3.5 generates recognition questions from captions, e.g., "What animal is shown in the image?"
>     - **Textual-centric Question Formulation:** GPT-3.5 generates diverse questions related to the visual content, e.g., "What is typical behavior of the rabbit?" along with answer options grounded in the visual context.
>     - **Evaluation and Filtering**: Questions are filtered based on LVLM performance, retaining only "easy" questions.
>
>     **Simple-Composed Question Generation**
>
>     - **Image Substitution**: Textual-centric questions are converted into Simple-Composed Questions by replacing text with images.
>     - **Filtering and Refinement**: Difficult questions are identified through LVLM performance and manually refined for the final benchmark.
>
>     We will include a **visual pipeline diagram** to make this process more transparent and easier to understand.
>
>
> > **W2 & Limitations 1.2:** Evaluation on the current mainstream mllms.
> >
>
> We include evaluations on several mainstream models, including open-source (Qwen2VL) and closed-source models (GPT-4.1, Gemini 2.0, Flash). As shown in the table below, these models demonstrate high accuracy on both visual- and textual-centric questions individually. However, when combined, they exhibit noticeable drop in performance (over 10%) compared to the individual task accuracies.
>
> This performance drop actually supports our motivation and quality of benchmark. Current mainstream MLLMs still show room for improvement on our VLComp benchmark, highlighting the quality and challenge of our benchmark.
>
> To further validate our benchmark, we also evaluated three human experts (with access to search engines) and report their average score. Their consistently high performance demonstrates that the benchmark is solvable by humans and confirms its clarity and diagnostic value.
>
> |  | Visual-centric questions Acc | Textual-centric questions Acc | Compositional questions Acc | $\Delta$ Performance Drop |
> | --- | --- | --- | --- | --- |
> | Human Expert | 100.0 | 95.1 | 94.1 | 1.0 |
> | *Closed Source Models* |  |  |  |  |
> | GPT-4o | 95.0 | 93.8 | 79.0 | 14.8 |
> | Gemini 2.0, Flash | 97.3 | 90.4 | 80.0 | 10.4 |
> | *Open Source Models* |  |  |  |  |
> | Qwen2VL | 96.4 | 78.1 | 67.3 | 10.8 |
>
> > **W3 & Limitations 2.2:** In the method part, the mllms used is a bit outdated
> >
>
> Thank you for the suggestion. We apply our method on mainstream Qwen2VL and evaluate them on VLComp. The results are as follows:
>
> |  | Perception | Science | Commonsense | Factual | Text | Scene | Math | Overall |
> | --- | --- | --- | --- | --- | --- | --- | --- | --- |
> | Qwen2VL | 74.5 | 71.7 | 62.5 | 69.0 | 60.7 | 63.9 | 66.7 | 67.3 |
> | + ours | 76.4 | 73.9 | 67.5 | 69.0 | 64.3 | 66.7 | 68.5 | 69.8 |
>
> The results indicate that our method also achieves the improvement on the mainstream baseline, which exhibits fewer hallucinations. Due to time constraints, we use only a very limited amount of training data from QwenVL (previous version in Qwen Family).
>
> > **Q1.1 & 2.1:** Discussion with multimodal alignment problem
> >
>
> Thank you for the question. SCHall arise not only from insufficient multimodal alignment, but also from failures in compositionality.  To further probe the role of modality alignment, we conduct an additional experiment using Qwen2VL. Specifically, we first prompt the model to generate a caption for the image alone, then combine this caption with the original textual question for inference. Interestingly, the model’s performance drop from **67.3% to 62.2%.** Among these, 8.3% of cases show improvement, indicating that alignment is indeed a bottleneck. However, the majority of new errors fall into two categories:  (1) 45.5% where the caption lack question-relevant visual details (reflecting visual abstraction failures, as discussed in Section 3.2), and (2) 54.5% where the caption is informative, but reasoning still failed—highlighting limits in compositional reasoning. These findings support our claim that **multimodal alignment, while necessary, is not sufficient to resolve SCHall cases**. Visual abstraction and other underlying factors remain critical challenges.
>
> Our proposed method also contributes to improving alignment. Among the 8.3% alignment-limited cases, our method improves 22%, demonstrating its effectiveness in addressing this subset of failures.
>
> > **Q1.2:** From Figure 1, the effect of simple-text is lower than that of simple-visual. Is there a corresponding explanation?
> >
>
> Since most existing hallucination benchmarks focus on visual recognition, we intentionally craft visually simple questions to minimize the impact of recognition errors and better diagnose SCHall.
>
> > **Q2.2 & Limitations 2.1:** Motivations and effects of the training stages in the method part.
> >
>
> The motivations and effects of two stages is as below:
>
> **Training Stage 1: VLR pretraining.** **Motivation:** As discussed in Section 3.2, a key contributor to SCHall is the failure of visual abstraction. To address this, we introduce explicit VLRs to encourage the model to abstract visual information effectively. Specifically, we pretrain the model by blocking access from the output tokens to the image, encouraging it to rely entirely on VLRs for extracting and transforming key information. **Effect:** This pretraining improves the model’s ability to extract and represent visual information accurately. However, it does not yet equip the model to jointly reason with images and VLRs, which is addressed in subsequent training stages.
>
> **Training Stage 2: Distillation learning.** **Motivation:** As shown in Section 3.3, the presence of visual inputs can degrade the model’s language processing ability. To address this issue, we distill language reasoning knowledge from a branch that uses only image captions to guide the main branch, which jointly reasons over images and VLRs. **Effect:** This stage improves the model’s language understanding in multimodal settings and encourages more effective use of VLRs.
>
> Thank you for the feedback. We will make the motivations and effects of each training stage clearer in future revisions of the paper.

---

> > ### Author Response · Authors · 2025-08-07
> >
> > Dear Reviewer r87F,
> >
> > As the discussion period is approaching its end in 2 days, we would greatly appreciate any further comments or suggestions to see if our responses have resolved your concerns.
> >
> > Thank you very much for your time and consideration.
> >
> > Sincerely,
> > The Authors

---

### Comment · Area_Chair_9vwn · 2025-08-02
**Reviewer-Author Discussions**

Dear Reviewers,
﻿

Could you kindly review the authors’ rebuttal as well as the comments from your fellow reviewers, and share your thoughts on the authors’ responses? Many thanks.
﻿

Best regards,

AC

---

### Note · Authors · 2025-08-13

Dear ACs and all reviewers,

We sincerely thank the ACs for the oversight and guidance during the discussion, and all reviewers for the insightful feedback.

First, we reiterate our main contributions, as recognized by the reviewers:

- **New Problem Definition**: We are the **first** to identify a fundamental compositional hallucination (SCHall) phenomenon, recognized by reviewers as a new hallucination type (4hUj), a novel evaluation perspective (r87F), and a significant, previously overlooked issue (cKHX).
- **Systematic Benchmarking and Validating SCHall Generalizability**: We propose VLComp for systematic SCHall evaluation, and validate SCHall’s prevalence across 7 LVLMs on 3 general-purpose benchmarks and VLComp. Reviewers commend VLComp’s diversity and careful design (4hUj), its novelty as the first SCHall evaluation, and its clear differentiation from existing hallucination benchmarks (cKHX).
- **In-depth Analysis:** We thoroughly reveals two primary factors contributing to SCHall, which reviewers found thoughtful (riVB) and supported by compelling empirical evidence (4hUj, cKHX).
- **Novel Method with Strong Performance:** Our VLR-distillation approach achieves substantial improvements on **both VLComp and general VQA benchmarks.** Reviewers praise it as elegant, practically implementable (4hUj), and effective (r87F, riVB).
- **Clear Organization:** Reviewers (r87F, cKHX) acknowledge the clarity and logical organization of our paper.

Second, we have carefully addressed all key concerns:

- **Additional experiments to demonstrate generalizability of SCHall and the effectiveness of our method.** We extend our evaluation to mainstream MLLMs, showing consistent improvements on VLComp (r87F W2 & W3) and also across diverse baselines and benchmarks (4hUj W3). r87F and 4hUj acknowledge this.
- **Distinctiveness and Quality of VLComp.** We provide **both theoretical comparisons and detailed statistical analyses** distinguishing VLComp from OK-VQA (riVB W1). Furthermore, we clarify its effectiveness in diagnosing SCHall and its scalability (4hUj W1, riVB W2). 4hUj confirms the concern resolved; riVB agrees that prior VQA datasets don’t target compositional hallucinations and values a dedicated compositional evaluation.
- **Theoretical Justification.** We offer theoretical analysis of SCHall (cKHX W2), detailed justification for benchmark design and methodology (r87F W1 & Q2). cKHX expresses a positive view; r87F acknowledges them.

Sincerely, The Authors

---

### Decision · Program_Chairs · 2025-09-17

**Decision:**

Accept (poster)

**Comment:**

This paper was reviewed by four experts in the field. The paper received mixed reviews, i.e., 4 Borderline Accept, 4 Borderline Accept, 3 Borderline reject, 6 Strong Accept.
After the discussion, the reviewers still have some concerns regarding the paper, such as
(a) The key differences between the proposed benchmark and existing commonsense-based VQA datasets.
(b) Not clear on the distinction between hallucination and traditional VQA errors.
On the other hand, the reviewers acknowledged the paper for its strengths in
(1) The paper is clearly written and well-structured.
(2) The paper addresses a previously unrecognized and important phenomenon—Simple Compositional Hallucination (SCHall).
(3) The identification and formalization of SCHall is novel.
(4) The proposed Vision Language Registers (VLRs) and distillation learning strategy are elegant, practically implementable solutions.
Based on these positive reviews, the decision was made to recommend it for acceptance. We congratulate the authors on their acceptance!
Besides, authors should revise the paper taking into account the reviewers' comments.